# EVALUATION OF SIMILARITY-BASED EXPLANATIONS

**Kazuaki Hanawa**[1,2]**, Sho Yokoi**[2,1]**, Satoshi Hara**[3]**, Kentaro Inui**[2,1]

RIKEN Center for Advanced Intelligence Project[1], Tohoku University[2], Osaka University[3]

`kazuaki.hanawa@riken.jp, yokoi@ecei.tohoku.ac.jp,`
`satohara@ar.sanken.osaka-u.ac.jp, inui@ecei.tohoku.ac.jp`

## ABSTRACT

Explaining the predictions made by complex machine learning models helps users to understand and accept the predicted outputs with confidence. One promising way is to use similarity-based explanation that provides similar instances as evidence to support model predictions. Several *relevance metrics* are used for this purpose. In this study, we investigated relevance metrics that can provide reasonable explanations to users. Specifically, we adopted three tests to evaluate whether the relevance metrics satisfy the minimal requirements for similarity-based explanation. Our experiments revealed that the cosine similarity of the gradients of the loss performs best, which would be a recommended choice in practice. In addition, we showed that some metrics perform poorly in our tests and analyzed the reasons of their failure. We expect our insights to help practitioners in selecting appropriate relevance metrics and also aid further researches for designing better relevance metrics for explanations.

## 1 INTRODUCTION

Explaining the predictions made by complex machine learning models helps users understand and accept the predicted outputs with confidence (Ribeiro et al., 2016; Lundberg & Lee, 2017; Guidotti et al., 2018; Adadi & Berrada, 2018; Molnar, 2020). Instance-based explanations are a popular type of explanation that achieve this goal by presenting one or several training instances that support the predictions of a model. Several types of instance-based explanations have been proposed, such as explaining with instances similar to the instance of interest (i.e., the test instance in question) (Charpiat et al., 2019; Barshan et al., 2020); harmful instances that degrade the performance of models (Koh & Liang, 2017; Khanna et al., 2019); counter-examples that contrast how a prediction can be changed (Wachter et al., 2018); and irregular instances (Kim et al., 2016).

Among these, we focus on the first one, the type of explanation that gives one or several training instances that are similar to the test instance in question and corresponding model predictions. We refer to this type of instance-based explanation as *similarity-based explanation*. A similarity-based explanation is of the form "I (the model) think this image is cat because similar images I saw in the past were also cat." This type of explanation is analogous to the way humans make decisions by referring to their prior experiences (Klein & Calderwood, 1988; Klein, 1989; Read & Cesa, 1991). Hence, it tends to be easy to understand even to users with little expertise about machine learning. A report stated that with this type of explanation, users tend to have higher confidence in model predictions compared to explanations that presents contributing features (Cunningham et al., 2003).

In the instance-based explanation paradigm, including similarity-based explanation, a relevance metric $R(\boldsymbol{z}, \boldsymbol{z}') \in \mathbb{R}$ is typically used to quantify the relationship between two instances, $\boldsymbol{z} = (\boldsymbol{x}, y)$ and $\boldsymbol{z}' = (\boldsymbol{x}', y')$.

**Definition 1** (Instance-based Explanation Using Relevance Metric)**.** Let $\mathcal{D} = \{\boldsymbol{z}_{\text{train}}^{(i)} = (\boldsymbol{x}_{\text{train}}^{(i)}, y_{\text{train}}^{(i)})\}_{i=1}^{N}$ be a set of training instances and $\boldsymbol{x}_{\text{test}}$ be a test input of interest whose predicted output is given by $\widehat{y}_{\text{test}} = f(\boldsymbol{x}_{\text{test}})$ with a predictive model $f$. An instance-based explanation method gives the most relevant training instance $\bar{\boldsymbol{z}} \in \mathcal{D}$ to the test instance $\boldsymbol{z}_{\text{test}} = (\boldsymbol{x}_{\text{test}}, \widehat{y}_{\text{test}})$ by $\bar{\boldsymbol{z}} = \arg\max_{\boldsymbol{z}_{\text{train}} \in \mathcal{D}} R(\boldsymbol{z}_{\text{test}}, \boldsymbol{z}_{\text{train}})$ using a relevance metric $R(\boldsymbol{z}_{\text{test}}, \boldsymbol{z}_{\text{train}})$.

Previously proposed relevance metrics include *similarity* (Caruana et al., 1999), *kernel functions* (Kim et al., 2016; Khanna et al., 2019), and *influence function* (Koh & Liang, 2017).

Table 1: The relevance metrics and their evaluation results. For the model randomization test, the results that passed the test are colored. For the identical class test and identical subclass test, the results with the five highest average evaluation scores are colored. The details of the relevance metrics, the evaluation criteria, and the evaluation procedures can be found in Sections 1.2, 3, and 4, respectively.

| Relevance Metrics | | Abbrv. | Evaluation Criteria | | |
|---|---|---|---|---|---|
| | | | Model Randomization Test | Identical Class Test | Identical Subclass Test |
| $\ell_2$ | $\phi(z) = x$ | $\ell_2^x$ | Failed | $0.615 \pm 0.261$ | $0.644 \pm 0.264$ |
| | $\phi(z) = h^{\text{last}}$ | $\ell_2^{\text{last}}$ | Passed | $0.880 \pm 0.106$ | $0.631 \pm 0.237$ |
| | $\phi(z) = h^{\text{all}}$ | $\ell_2^{\text{all}}$ | Failed | $0.848 \pm 0.128$ | $0.691 \pm 0.211$ |
| Cosine | $\phi(z) = x$ | $\cos^x$ | Failed | $0.669 \pm 0.248$ | $0.621 \pm 0.242$ |
| | $\phi(z) = h^{\text{last}}$ | $\cos^{\text{last}}$ | Passed | $0.888 \pm 0.098$ | $0.636 \pm 0.234$ |
| | $\phi(z) = h^{\text{all}}$ | $\cos^{\text{all}}$ | Failed | $0.871 \pm 0.110$ | $0.738 \pm 0.166$ |
| Dot | $\phi(z) = x$ | $\text{dot}^x$ | Failed | $0.336 \pm 0.187$ | $0.346 \pm 0.201$ |
| | $\phi(z) = h^{\text{last}}$ | $\text{dot}^{\text{last}}$ | Failed | $0.579 \pm 0.344$ | $0.284 \pm 0.122$ |
| | $\phi(z) = h^{\text{all}}$ | $\text{dot}^{\text{all}}$ | Failed | $0.630 \pm 0.353$ | $0.488 \pm 0.267$ |
| Gradient | Influence Function | IF | Passed | $0.372 \pm 0.270$ | $0.309 \pm 0.174$ |
| | Relative IF | RIF | Passed | $0.779 \pm 0.309$ | $0.659 \pm 0.266$ |
| | Fisher Kernel | FK | Passed | $0.226 \pm 0.103$ | $0.180 \pm 0.076$ |
| | Grad-Dot | GD | Passed | $0.701 \pm 0.287$ | $0.403 \pm 0.131$ |
| | Grad-Cos | GC | Passed | $0.996 \pm 0.009$ | $0.753 \pm 0.196$ |

An immediate critical question is which relevance metric is appropriate for which type of instance-based explanations. There is no doubt that different types of explanations require different metrics. Despite its potential importance, however, little has been explored on this question. Given this background, in this study, we focused on similarity-based explanation and investigated its appropriate relevance metrics through comprehensive experiments.[1]

**Contributions** We provide the first answer to the question about which relevance metrics have desirable properties for similarity-based explanation. For this purpose, we propose to use three *minimal requirement* tests to evaluate various relevance metrics in terms of their appropriateness. The first test is the model randomization test originally proposed by Adebayo et al. (2018) for evaluating saliency-based methods, and the other two tests, the identical class test and identical subclass test, are newly designed in this study. As summarized in Table 1, our experiments revealed that (i) the cosine similarity of gradients performs best, which is probably a recommended choice for similarity-based explanation in practice, and (ii) some relevance metrics demonstrated poor performances on the identical class and identical subclass tests, indicating that their use should be deprecated for similarity-based explanation. We also analyzed the reasons behind the success and failure of metrics. We expect these insights to help practitioners in selecting appropriate relevance metrics.

## 1.1 PRELIMINARIES

**Notations** For vectors $a, b \in \mathbb{R}^p$, we denote the dot product by $\langle a, b \rangle := \sum_{i=1}^{p} a_i b_i$, the $\ell_2$ norm by $\|a\| := \sqrt{\langle a, a \rangle}$, and the cosine similarity by $\cos(a, b) := \langle a, b \rangle / \|a\| \|b\|$.

**Classification Problem** We consider a standard classification problem as the evaluation benchmark, which is the most actively explored application of instance-based explanations. The model is the conditional probability $p(y \mid x; \theta)$ with parameter $\theta$. Let $\widehat{\theta}$ be a trained parameter $\widehat{\theta} = \arg\min_{\theta} \mathcal{L}_{\text{train}} := \frac{1}{N} \sum_{i=1}^{N} \ell(z_{\text{train}}^{(i)}; \theta)$, where the loss function $\ell$ is the cross entropy $\ell(z; \theta) = -\log p(y \mid x; \theta)$ for an input-output pair $z = (x, y)$. The model classifies a test input $x_{\text{test}}$ by assigning the class with the highest probability $\widehat{y}_{\text{test}} = \arg\max_y p(y \mid x_{\text{test}}; \widehat{\theta})$.

## 1.2 RELEVANCE METRICS

We present an overview of the two types of relevance metrics considered in this study, namely *similarity metrics* and *gradient-based metrics*. To the best of our knowledge, all major relevance

---

[1]Our implementation is available at `https://github.com/k-hanawa/criteria_for_instance_based_explanation`

metrics proposed thus far can be classified under these two types. Table 1 presents a list of metrics and their abbreviations.

**Similarity Metrics** We consider the following popular similarity metrics with a feature map $\phi(\boldsymbol{z})$.

- **$\ell_2$ Metric**: $R_{\ell_2}(\boldsymbol{z}, \boldsymbol{z}') := -\|\phi(\boldsymbol{z}) - \phi(\boldsymbol{z}')\|^2$, which is a typical choice for nearest neighbor methods (Hastie et al., 2009; Abu Alfeilat et al., 2019).

- **Cosine Metric**: $R_{\cos}(\boldsymbol{z}, \boldsymbol{z}') := \cos(\phi(\boldsymbol{z}), \phi(\boldsymbol{z}'))$, which is commonly used in natural language processing tasks (Mikolov et al., 2013; Arora et al., 2017; Conneau et al., 2017).

- **Dot Metric**: $R_{\mathrm{dot}}(\boldsymbol{z}, \boldsymbol{z}') := \langle \phi(\boldsymbol{z}), \phi(\boldsymbol{z}') \rangle$, which is a kernel function used in kernel models such as SVM (Schölkopf et al., 2002; Fan et al., 2005; Bien & Tibshirani, 2011).

As the feature map $\phi(\boldsymbol{z})$, we consider (i) an identity map $\phi(\boldsymbol{z}) = \boldsymbol{x}$; (ii) the last hidden layer $\phi(\boldsymbol{z}) = \boldsymbol{h}^{\mathrm{last}}$, which is the latent representation of input $\boldsymbol{x}$, one layer before the output in a deep neural network; and, (iii) all hidden layers $\phi(\boldsymbol{z}) = \boldsymbol{h}^{\mathrm{all}}$, where $\boldsymbol{h}^{\mathrm{all}} = [\boldsymbol{h}^1, \boldsymbol{h}^2, \dots, \boldsymbol{h}^{\mathrm{last}}]$ is the concatenation of all latent representations in the network. Note that the metrics with the identity map merely measure the similarity of inputs without model information. We adopt these metrics as naive baselines to contrast with other advanced metrics that utilize model information.

**Gradient-based Metrics** Gradient-based metrics use a gradient $\boldsymbol{g}_{\widehat{\boldsymbol{\theta}}}^{\boldsymbol{z}} := \nabla_{\boldsymbol{\theta}} \ell(\boldsymbol{z}; \widehat{\boldsymbol{\theta}})$ to measure the relevance. We consider five metrics: Influence Function (IF) (Koh & Liang, 2017), Relative IF (RIF) (Barshan et al., 2020), Fisher Kernel (FK) (Khanna et al., 2019), Grad-Dot (GD) (Yeh et al., 2018; Charpiat et al., 2019), and Grad-Cos (GC) (Perronnin et al., 2010; Charpiat et al., 2019). See Appendix A for further detail.

- **IF**: $R_{\mathrm{IF}}(\boldsymbol{z}, \boldsymbol{z}') := \langle \boldsymbol{g}_{\widehat{\boldsymbol{\theta}}}^{\boldsymbol{z}}, \boldsymbol{H}^{-1} \boldsymbol{g}_{\widehat{\boldsymbol{\theta}}}^{\boldsymbol{z}'} \rangle$

- **RIF**: $R_{\mathrm{RIF}}(\boldsymbol{z}, \boldsymbol{z}') := \cos(\boldsymbol{H}^{-\frac{1}{2}} \boldsymbol{g}_{\widehat{\boldsymbol{\theta}}}^{\boldsymbol{z}}, \boldsymbol{H}^{-\frac{1}{2}} \boldsymbol{g}_{\widehat{\boldsymbol{\theta}}}^{\boldsymbol{z}'})$

- **FK**: $R_{\mathrm{FK}}(\boldsymbol{z}, \boldsymbol{z}') := \langle \boldsymbol{g}_{\widehat{\boldsymbol{\theta}}}^{\boldsymbol{z}}, \boldsymbol{I}^{-1} \boldsymbol{g}_{\widehat{\boldsymbol{\theta}}}^{\boldsymbol{z}'} \rangle$,

- **GD**: $R_{\mathrm{GD}}(\boldsymbol{z}, \boldsymbol{z}') := \langle \boldsymbol{g}_{\widehat{\boldsymbol{\theta}}}^{\boldsymbol{z}}, \boldsymbol{g}_{\widehat{\boldsymbol{\theta}}}^{\boldsymbol{z}'} \rangle$

- **GC**: $R_{\mathrm{GC}}(\boldsymbol{z}, \boldsymbol{z}') := \cos(\boldsymbol{g}_{\widehat{\boldsymbol{\theta}}}^{\boldsymbol{z}}, \boldsymbol{g}_{\widehat{\boldsymbol{\theta}}}^{\boldsymbol{z}'})$

where $\boldsymbol{H}$ and $\boldsymbol{I}$ are the Hessian and Fisher information matrices of the loss $\mathcal{L}_{\mathrm{train}}$, respectively.

## 2 RELATED WORK

**Model-specific Explanation** Aside of the relevance metrics, there is another approach for similarity-based explanation that uses specific models that can provide explanations by their design (Kim et al., 2014; Plötz & Roth, 2018; Chen et al., 2019). We set aside these specific models and focus on generic relevance metrics because of their applicability to a wide range of problems.

**Evaluation of Metrics for Improving Classification Accuracy** In several machine learning problems, the metrics between instances play an essential role. For example, the distance between instances is essential for distance-based methods such as nearest neighbor methods (Hastie et al., 2009). Another example is kernel models where the kernel function represents the relationship between two instances (Schölkopf et al., 2002). Several studies have evaluated the desirable metrics for specific tasks (Hussain et al., 2011; Hu et al., 2016; Li & Li, 2018; Abu Alfeilat et al., 2019). These studies aimed to find metrics that could improve the classification accuracy. Different from these evaluations based on accuracy, our goal in this study is to evaluate the validity of relevance metrics for similarity-based explanation; thus, the findings in these previous studies are not directly applicable to our goal.

**Evaluation of Explanations** There are a variety of desiderata argued as requirements for explanations, such as faithfulness (Adebayo et al., 2018; Lakkaraju et al., 2019; Jacovi & Goldberg, 2020), plausibility (Lei et al., 2016; Lage et al., 2019; Strout et al., 2019), robustness (Alvarez-Melis & Jaakkola, 2018), and readability (Wang & Rudin, 2015; Yang et al., 2017; Angelino et al., 2017). It is important to evaluate the existing explanation methods considering these requirements. However, there is no standard test established for evaluating these requirements, and designing such tests still remains an open problem (Doshi-Velez & Kim, 2017; Jacovi & Goldberg, 2020). In this study, as the first empirical study for evaluating the existing relevance metrics for similarity-based explanation, we take an alternative approach by designing *minimal* requirement tests for two primary requirements,

namely faithfulness and plausibility. With this alternative approach, we can avoid the difficulty of directly evaluating these primary requirements.

## 3    EVALUATION CRITERIA FOR SIMILARITY-BASED EXPLANATION

This study aims to investigate the relevance metrics with desirable properties for similarity-based explanation. In this section, we propose three tests to evaluate whether the relevance metrics satisfy the minimal requirements for similarity-based explanation. If a relevance metric fails one of the tests, we can conclude that the metric does not meet the minimal requirements; thus, its use would be deprecated. The first test (model randomization test) assesses whether each relevance metric satisfies the minimal requirements for the *faithfulness* of explanation, which requires that an explanation to a model prediction must reflect the underlying inference process (Adebayo et al., 2018; Lakkaraju et al., 2019; Jacovi & Goldberg, 2020). The latter two tests (identical class and identical subclass tests) are designed to assess relevance metrics in terms of the *plausibility* of the explanations they produce (Lei et al., 2016; Lage et al., 2019; Strout et al., 2019), which requires explanations to be sufficiently convincing to users.

### 3.1    MODEL RANDOMIZATION TEST

Explanations that are irrelevant to a model should be avoided because such fake explanations can mislead users. Thus, any valid relevance metric should be model-dependent, which constitutes the first requirement.

We use the model randomization test of Adebayo et al. (2018) to assess whether a given relevance metric satisfies a minimal requirement for faithfulness. If a relevance metric produces almost same explanations for the same inputs on two models with different inference processes, it is likely to ignore the underlying model, i.e., the metric is independent of the model. Thus, we can evaluate whether the metric is model-dependent by comparing explanations from two different models. In the test, a typical choice of the models is a well-trained model that can predict the output well and a randomly initialized model that can make only poor prediction. These two models have different inference processes; hence, their explanations should be different.

**Definition 2** (Model Randomization Test). Let $R$ denote the relevance metric of interest. Let $f$ and $f_{\mathrm{rand}}$ be a well-trained model and randomly initialized model, respectively. For given $R$, $f$, and test instance $\boldsymbol{z}_{\mathrm{test}} = (\boldsymbol{x}_{\mathrm{test}}, \widehat{y}_{\mathrm{test}})$, let $\pi_f$ be a permutation of the indices of the training instances based on the degree of relevance to the given test instance, i.e., $R(\boldsymbol{z}_{\mathrm{test}}, \boldsymbol{z}_{\mathrm{train}}^{(\pi_f(1))}) \geq R(\boldsymbol{z}_{\mathrm{test}}, \boldsymbol{z}_{\mathrm{train}}^{(\pi_f(2))}) \geq \ldots \geq R(\boldsymbol{z}_{\mathrm{test}}, \boldsymbol{z}_{\mathrm{train}}^{(\pi_f(N))})$. We also define $\pi_{f_{\mathrm{rand}}}$ accordingly. Then, we require $\pi_f$ and $\pi_{f_{\mathrm{rand}}}$ to ensure a small rank correlation.

If relevance metric $R$ is independent of the model, it produces the same permutation for both $f$ and $f_{\mathrm{rand}}$, and their rank correlation becomes one. If the rank correlation is significantly smaller than one and close to zero, we can confirm that the relevance metric is model-dependent.

### 3.2    IDENTICAL CLASS TEST

The second minimal requirement is that the raised similar instance should belong to the same class as the test instance, as shown in Figure 1. The violation of this requirement leads to nonsensical explanations such as "I think this image is cat because a similar image I saw in the past was dog." in Figure 1. When users encounter such explanations, they might question the validity of model predictions and ignore the predictions even if the underlying model is valid. This observation leads to the identical class test below.

**Definition 3** (Identical Class Test). We require that the most similar (relevant) instance of a test instance $\boldsymbol{z}_{\mathrm{test}} = (\boldsymbol{x}_{\mathrm{test}}, \widehat{y}_{\mathrm{test}})$ is a training instance of the same class as the given test instance.

$$\underset{\boldsymbol{z} = (\boldsymbol{x}, y) \in \mathcal{D}}{\arg\max} \; R(\boldsymbol{z}_{\mathrm{test}}, \boldsymbol{z}) = (\bar{\boldsymbol{x}}, \bar{y}) \implies \bar{y} = \widehat{y}_{\mathrm{test}}. \tag{1}$$

Although this test may look trivial, some relevance metrics do not satisfy this minimal requirement, as demonstrated in Section 4.2.

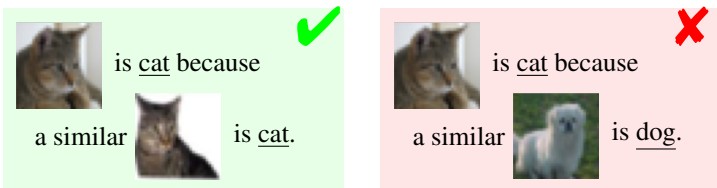

Figure 1: Valid (✔) and invalid (✘) examples for the *identical class test*.

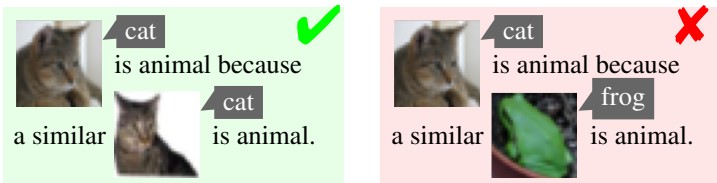

Figure 2: Valid (✔) and invalid (✘) examples for the *identical subclass test*.

## 3.3 IDENTICAL SUBCLASS TEST

The third minimal requirement is that the raised similar instance should belong to the same subclass as that of the test instance when the the classes consist of latent subclasses, as shown in Figure 2. For example, consider a problem of classifying images of CIFAR10 into two classes, i.e., animal and vehicle. The animal class consists of images from subclasses such as cat and frog, while the vehicle class consists of images from subclasses such as airplane and automobile. Under the presence of subclasses, the violation of this requirement leads to nonsensical explanations such as "I think this image (cat) is animal because a similar image (frog) I saw in the past was also animal." in Figure 2. This observation leads to the identical subclass test below.

**Definition 4** (Identical Subclass Test). Let $s(z)$ be a subclass for class $y$ of an instance $z = (x, y)$. We require that the most similar (relevant) instance of a test instance $z_{\text{test}} = (x_{\text{test}}, \widehat{y}_{\text{test}})$ is the training instance of the same subclass as the test instance, under the assumption that the prediction of the test instance is correct $\widehat{y}_{\text{test}} = y_{\text{test}}$.[2]

$$\arg\max_{z \in \mathcal{D}} R(z_{\text{test}}, z) = \bar{z} \implies s(bar z) = s(z_{\text{test}}). \tag{2}$$

In the experiments, we used modified datasets: we split the dataset into two new classes (A and B) by randomly assigning the existing classes to either classes. The new two classes now contain the original data classes as subclasses that are mutually exclusive and collectively exhaustive, which can be used for the identical subclass test.

## 3.4 DISCUSSIONS ON VALIDITY OF CRITERIA

Here, we discuss the validity of the new criteria, i.e., the identical class and identical subclass tests.

**Why do relevance metrics that cannot pass these tests matter?** Dietvorst et al. (2015) revealed a bias in humans, called algorithm aversion, which states that people tend to ignore an algorithm if it makes errors. It should be noted that the explanations that do not satisfy the identical class test or identical subclass test appear to be logically broken, as shown in Figures 1 and 2. Given such logically broken explanations, users will consider that the models are making errors, even if they are making accurate predictions. Eventually, the users will start to ignore the models.

**Is the identical subclass test necessary?** This is an essential requirement for ensuring that the explanations are plausible to *any* users. Some users may not consider the explanations that violate the identical subclass test to be logically broken. For example, some users may find a frog to be an appropriate explanation for a cat being animal by inferring taxonomy of the classes (e.g., both have eyes). However, we cannot hope all users to infer the same taxonomy. Therefore, if there is

---

[2]We require correct predictions in this test because the subclass does not match for incorrect cases.

a discrepancy between the explanation and the taxonomy inferred by a user, the user will consider the explanation to be implausible. To make explanations plausible to any user, instances of the same subclass need to be provided.

**Is random class assignment in the identical subclass test appropriate?** We adopted random assignment to evaluate the performance of each metric independent from the underlying taxonomy. If a specific taxonomy was considered for the evaluations, a metric that performed well with it will be highly valued. Random assignment eliminates such effects, and we can purely measure the performance of the metrics themselves.

**Do classification models actually recognize subclasses? Is the identical subclass test suitable to evaluate the explanations of predictions *made by practical models?*** It is true that if a model ignores subclasses in its training and inference processes, any explanation will fail the test. We conducted simple preliminary experiments and confirmed that the practical classification models used in this study capture the subclasses. See Appendix E for further detail.

## 4 EVALUATION RESULTS

Here, we examine the validity of relevance metrics with respect to the three minimal requirements. For this evaluation, we used two image datasets (MNIST (LeCun et al., 1998), CIFAR10 (Krizhevsky, 2009)), two text datasets (TREC (Li & Roth, 2002), AGNews (Zhang et al., 2015)) and two table datasets (Vehicle (Dua & Graff, 2017), Segment (Dua & Graff, 2017)). As benchmarks, we employed logistic regression and deep neural networks trained on these datasets. Details of the datasets, models, and computing infrastructure used in this study is provided in Appendix B.

**Procedure** We repeated the following procedure 10 times for each evaluation test.

1. Train a model using a subset of training instances.[3] Then, randomly sample $500$ test instances from the test set.[4]

2. For each test instance, compute the relevance score for all instances used for training.

3. (a) For the model randomization test, compute the Spearman rank correlation coefficients between the relevance scores from the trained model and relevance scores from the randomized model.

   (b) For the identical class and identical subclass tests, compute the success rate, which is the ratio of test instances that passed the test.

In this section, we mainly present the results for CIFAR10 with CNN and AGNews with Bi-LSTM. The other results were similar, and can be found in Appendix F.

**Result Summary** We summarize the main results before discussing individual results.

- $\ell_2^{\text{last}}$, $\cos^{\text{last}}$, and gradient-based metrics scored low correlation in the model randomization test for all datasets and models, indicating that they are model-dependent.

- GC performed the best in most of the identical class and identical subclass tests; thus, GC would be the recommended choice in practice.

- Dot metrics as well as IF, FK, and GD performed poorly on the identical class test and identical subclass test.

In Section 5, we analyze why some relevance metrics succeed or fail in the identical class and identical subclass tests.

### 4.1 RESULT OF MODEL RANDOMIZATION TEST

Figure 3 shows the Spearman rank correlation coefficients for the model randomization test. The similarities with the identity feature map $\ell_2^x$, $\cos^x$, and $\text{dot}^x$ are irrelevant to the model and their correlations are trivially one. In the figures, the other metrics scored correlations close to zero,

---

[3]We randomly sampled 10% of MNIST and CIFAR10; 50% of TREC, Vehicle and Segment; and 5% of AGNews

[4]For the identical subclass test, we sampled instances with correct predictions only.

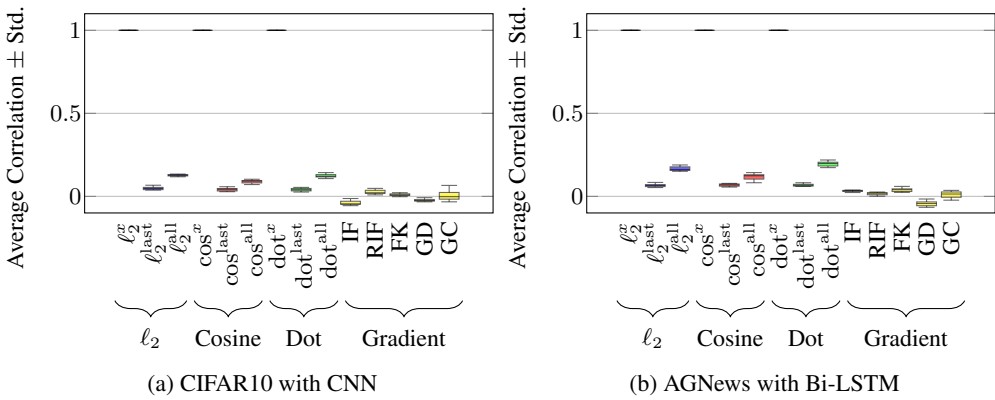

Figure 3: Result of the model randomization test. Correlations close to zero are ideal.

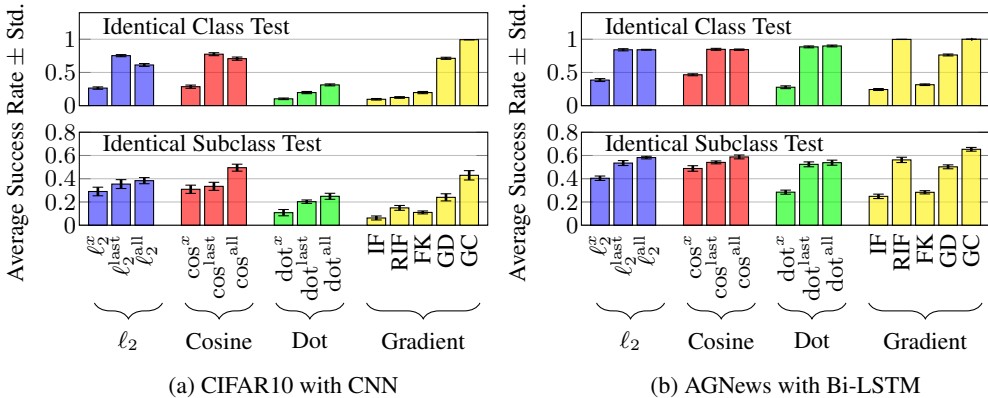

Figure 4: Results of the identical class test and identical subclass test.

indicating they will be model-dependent. However, the correlation of $\ell_2^{\text{all}}$, $\cos^{\text{all}}$, $\text{dot}^{\text{last}}$ was observed to be more than 0.7 on the MNIST and Vehicle datasets (see Appendix F). Therefore, we conclude that these relevance metrics failed the model randomization test because they can raise instances irrelevant to the model for some datasets.

## 4.2 RESULTS OF IDENTICAL CLASS AND IDENTICAL SUBCLASS TESTS

Figure 4 depicts the success rates for the identical class and identical subclass tests. We also summarized the average success rates of our experiments in Table 1. It is noteworthy that GC performed consistently well on the identical class and identical subclass tests for all the datasets and models used in the experiment (see Appendix F). In contrast, some relevance metrics such as the dot metrics as well as IF, FK, and GD performed poorly on both tests. The reasons for their failure are discussed in the next section.

To conclude, the results of our evaluations indicate that only GC performed well on all tests. That is, only GC seems to meet the minimal requirements; thus, it would be a recommended choice for similarity-based explanation.

## 5 WHY SOME METRICS ARE SUCCESSFUL AND WHY SOME ARE NOT

We observed that the dot metrics and gradient-based metrics such as IF, FK, and GD failed the identical class and identical subclass tests, in comparison to GC that exhibited remarkable performance. Here, we analyze the reasons why the aforementioned metrics failed while GC performed well. In Appendix D, we also discuss a way to *repair* IF, FK, and GD to improve their performance based on the findings in this section.

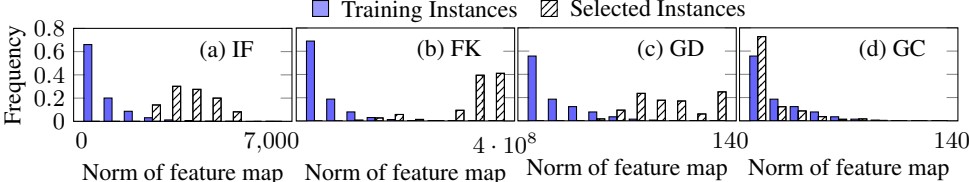

Figure 5: Distributions of norms of the feature maps of all training instances (colored) and the instances selected by the identical class test (meshed) on CIFAR10 with CNN.

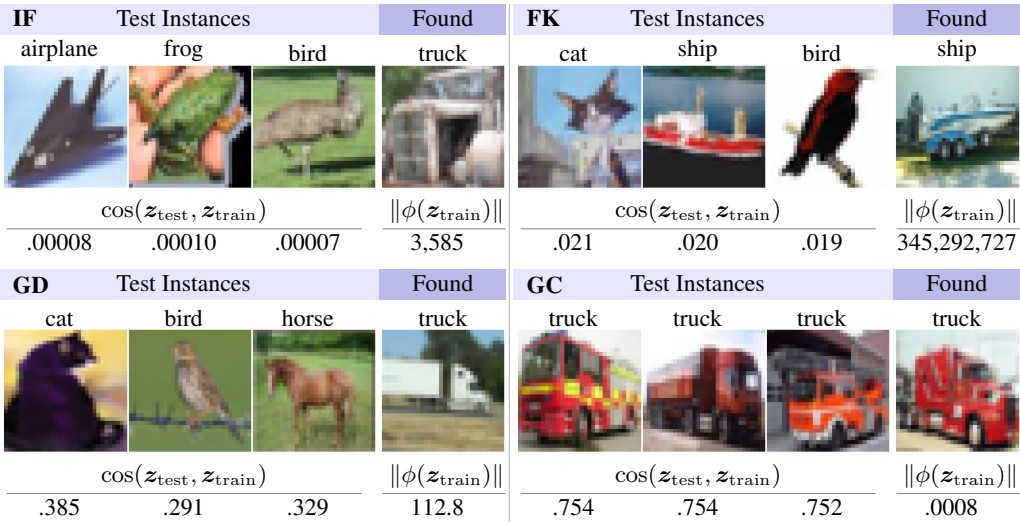

Figure 6: Training instances frequently selected in the identical class test with multiple test instances on CIFAR10 with CNN, the cosine between them, and the norm of training instances.

**Failure of Dot Metrics and Gradient-based Metrics**    To understand the failure, we reformulate IF, FK, and GD as dot metrics of the form $R_{\text{dot}}(z_{\text{test}}, z_{\text{train}}) = \langle \phi(z_{\text{test}}), \phi(z_{\text{train}}) \rangle$ to ensure that the following discussion is valid for any relevance metric of this form. It is evident that IF, FK, and GD can be expressed in this form by defining the feature maps by $\phi(z) = H^{-1/2}g(z; \widehat{\theta})$, $\phi(z) = I^{-1/2}g(z; \widehat{\theta})$, and $\phi(z) = g(z; \widehat{\theta})$, respectively.

Given a criterion, let $z_{\text{train}}^{(i)}$ be a desirable instance for a test instance $z_{\text{test}}$. The failures of dot metrics indicate the existence of an undesirable instance $z_{\text{train}}^{(j)}$ such that $\langle \phi(z_{\text{test}}), \phi(z_{\text{train}}^{(i)}) \rangle < \langle \phi(z_{\text{test}}), \phi(z_{\text{train}}^{(j)}) \rangle$. The following sufficient condition for $z_{\text{train}}^{(j)}$ is useful to understand the failure.

$$\|\phi(z_{\text{train}}^{(i)})\| < \|\phi(z_{\text{train}}^{(j)})\| \cos(\phi(z_{\text{test}}), \phi(z_{\text{train}}^{(j)})). \tag{3}$$

The condition implies that any instance with an extremely large norm and a cosine slightly larger than zero can be the candidate of $z_{\text{train}}^{(j)}$. In our experiments, we observed that the condition on the norm is especially crucial. As shown in Figure 5, even though instances with significanty large norms were scarce, only such extreme instances were selected as relevant instances by IF, FK, and GD. This indicates that these these metrics tend to consider such extreme instances as relevant. In contrast, GC was not attracted by large norms because it completely cancels the norm through normalization.

Figure 6 shows some training instances frequently selected in the identical class test on CIFAR10 with CNN. When using IF, FK, and GD, these training instances were frequently selected irrespective of their classes because the training instances had large norms. In these metrics, the term $\cos(\phi(z_{\text{test}}), \phi(z_{\text{train}}))$ seems to have negligible effects. In contrast, GC successfully selected the instances of the same class and ignored those with large norms.

**Success of GC**    We now analyze why GC performed well, specifically in the identical class test. To simplify the discussion, we consider linear logistic regression whose conditional distribution $p(y \mid x; \theta)$ is given by the $y$-th entry of $\sigma(Wx)$, where $\sigma$ is the softmax function, $\theta = W \in \mathbb{R}^{C \times d}$,

and $C$ and $d$ denote the number of classes and dimensionality of $\boldsymbol{x}$, respectively. With some algebra, we obtain $R_{\mathrm{GC}}(\boldsymbol{z}, \boldsymbol{z}') = \cos(\boldsymbol{r^z}, \boldsymbol{r^{z'}}) \cos(\boldsymbol{x}, \boldsymbol{x}')$ for $\boldsymbol{z} = (\boldsymbol{x}, y)$ and $\boldsymbol{z}' = (\boldsymbol{x}', y')$, where $\boldsymbol{r^z} = \sigma(W\boldsymbol{x}) - \boldsymbol{e}_y$ is the *residual* for the prediction on $\boldsymbol{z}$ and $\boldsymbol{e}_y$ is a vector whose $y$-th entry is one, and zero, otherwise. See Appendix C for the derivation. Here, the term $\cos(\boldsymbol{r^z}, \boldsymbol{r^{z'}})$ plays an essential role in GC. By definition, $r_c^{\boldsymbol{z}} \le 0$ if $c = y$ and $r_c^{\boldsymbol{z}} \ge 0$, otherwise. Thus, $\cos(\boldsymbol{r^z}, \boldsymbol{r^{z'}}) \ge 0$ always holds true when $y = y'$, while $\cos(\boldsymbol{r^z}, \boldsymbol{r^{z'}})$ can be negative for $y \ne y'$. Hence, the chance of $R_{\mathrm{GC}}(\boldsymbol{z}, \boldsymbol{z}')$ being positive can be larger for the instances from the same class compared to those from a different class.

Figure 7 shows that $\cos(\boldsymbol{r^z}, \boldsymbol{r^{z'}})$ is essential also for deep neural networks. Here, for each test instance $\boldsymbol{z}_{\mathrm{test}}$ on CIFAR10 with CNN, we randomly sampled two training instances $\boldsymbol{z}_{\mathrm{train}}$ (one with the same class and the other with a different class), and computed $R_{\mathrm{GC}}(\boldsymbol{z}_{\mathrm{test}}, \boldsymbol{z}_{\mathrm{train}})$ and $\cos(\boldsymbol{r^{z_{\mathrm{test}}}}, \boldsymbol{r^{z_{\mathrm{train}}}})$.

We also note that $\cos(\boldsymbol{r^{z_{\mathrm{test}}}}, \boldsymbol{r^{z_{\mathrm{train}}}})$ alone was not helpful for the identical subclass test, whose success rate was around the chance level. We thus conjecture that while $\cos(\boldsymbol{r^{z_{\mathrm{test}}}}, \boldsymbol{r^{z_{\mathrm{train}}}})$ is particularly helpful for the identical class test, the use of the entire gradient is still essential for GC to work effectively.

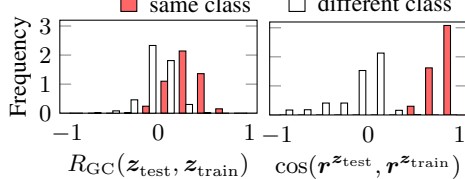

Figure 7: Distributions of $R_{\mathrm{GC}}(\boldsymbol{z}_{\mathrm{test}}, \boldsymbol{z}_{\mathrm{train}})$ and $\cos(\boldsymbol{r^{z_{\mathrm{test}}}}, \boldsymbol{r^{z_{\mathrm{train}}}})$ for training instances with the same / different classes on CIFAR10 with CNN.

## 6   CONCLUSION

We investigated and determined relevance metrics that are effective for similarity-based explanation. For this purpose, we evaluated whether the metrics satisfied the minimal requirements for similarity-based explanation. In this study, we conducted three tests, namely, the model randomization test of Adebayo et al. (2018) to evaluate whether the metrics are model-dependent, and two newly designed tests, the identical class and identical subclass tests, to evaluate whether the metrics can provide plausible explanations. Quantitative evaluations based on these tests revealed that the cosine similarity of gradients performs best, which would be a recommended choice in practice. We also observed that some relevance metrics do not meet the requirements; thus, the use of such metrics would not be appropriate for similarity-based explanation. We expect our insights to help practitioners in selecting appropriate relevance metrics, and also to help further researches for designing better relevance metrics for instance-based explanations.

Finally, we present two future direction for this study. First, the proposed criteria only evaluated limited aspects of the faithfulness and plausibility of relevance metrics. Thus, it is important to investigate further criteria for more detailed evaluations. Second, in addition to similarity-based explanation, it is necessary to consider the evaluation of other explanation methods, such as counter-examples. We expect this study to be the first step toward the rigorous evaluation of several instance-based explanation methods.

### ACKNOWLEDGMENTS

We thank Dr. Ryo Karakida and Dr. Takanori Maehara for their helpful advice. We also thank Overfit Summer Seminar[5] for an opportunity that inspired this research. Additionally, we are grateful to our laboratory members for their helpful comments. Sho Yokoi was supported by JST, ACT-X Grant Number JPMJAX200S, Japan. Satoshi Hara was supported by JSPS KAKENHI Grant Number 20K19860, and JST, PRESTO Grant Number JPMJPR20C8, Japan.

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

## A GRADIENT-BASED METRICS

In gradient-based metrics, we consider a model with parameter $\boldsymbol{\theta}$, its loss $\ell(\boldsymbol{z}; \boldsymbol{\theta})$, and its gradient $\nabla_{\boldsymbol{\theta}}\ell(\boldsymbol{z}; \boldsymbol{\theta})$ to measure relevance, where $\boldsymbol{z} = (\boldsymbol{x}, y)$ is an input-output pair.

**Influence Function (Koh & Liang, 2017)**  Koh & Liang (2017) proposed to measure relevance according to "how largely the test loss will increase if the training instance is omitted from the training set." Here, the model parameter trained using all of the training set is denoted by $\widehat{\boldsymbol{\theta}}$, and the parameter trained using all of the training set except the $i$-th instance $\boldsymbol{z}_{\text{train}}^{(i)}$ is denoted by $\widehat{\boldsymbol{\theta}}_{-i}$. The relevance metric proposed by Koh & Liang (2017) is then defined as the difference between the test loss under parameters $\widehat{\boldsymbol{\theta}}$ and $\widehat{\boldsymbol{\theta}}_{-i}$ as follows:

$$R_{\text{IF}}(\boldsymbol{z}_{\text{test}}, \boldsymbol{z}_{\text{train}}^{(i)}) := \ell(\boldsymbol{z}_{\text{test}}; \widehat{\boldsymbol{\theta}}_{-i}) - \ell(\boldsymbol{z}_{\text{test}}; \widehat{\boldsymbol{\theta}}). \tag{4}$$

Here, a greater value indicates that the loss on the test instance increases drastically by removing the $i$-th training instance from the training set. Thus, the $i$-th training instance is essential relative to predicting the test instance; therefore, it is highly relevant.

In practice, the following approximation is used to avoid computing $\widehat{\boldsymbol{\theta}}_{-i}$ explicitly.

$$R_{\text{IF}}(\boldsymbol{z}_{\text{test}}, \boldsymbol{z}_{\text{train}}^{(i)}) \approx \langle \nabla_{\boldsymbol{\theta}}\ell(\boldsymbol{z}_{\text{test}}; \widehat{\boldsymbol{\theta}}), \boldsymbol{H}^{-1}\nabla_{\boldsymbol{\theta}}\ell(\boldsymbol{z}_{\text{train}}^{(i)}; \widehat{\boldsymbol{\theta}})) \rangle, \tag{5}$$

where $\boldsymbol{H}$ is the Hessian matrix of the loss $\mathcal{L}_{\text{train}}$.

**Relative IF (Barshan et al., 2020)**  Barshan et al. (2020) proposed to measure relevance according to "how largely the test loss will increase if the training instance is omitted from the training set under the constraint that the expected squared change in loss is sufficiently small"[6], which is the modified version of the influence function. Relative IF is computed as the cosine similarity of $\phi(\boldsymbol{z}) = \boldsymbol{H}^{-1/2}\nabla_{\boldsymbol{\theta}}\ell(\boldsymbol{z}; \widehat{\boldsymbol{\theta}})$:

$$R_{\text{RIF}}(\boldsymbol{z}_{\text{test}}, \boldsymbol{z}_{\text{train}}) := \cos(\boldsymbol{H}^{-1/2}\nabla_{\boldsymbol{\theta}}\ell(\boldsymbol{z}_{\text{test}}; \widehat{\boldsymbol{\theta}}), \boldsymbol{H}^{-1/2}\nabla_{\boldsymbol{\theta}}\ell(\boldsymbol{z}_{\text{train}}; \widehat{\boldsymbol{\theta}})). \tag{6}$$

**Fisher Kernel (Khanna et al., 2019)**  Khanna et al. (2019) proposed to measure the relevance of instances using the Fisher kernel as follows:

$$R_{\text{FK}}(\boldsymbol{z}_{\text{test}}, \boldsymbol{z}_{\text{train}}^{(i)}) := \langle \nabla_{\boldsymbol{\theta}}\ell(\boldsymbol{z}_{\text{test}}; \widehat{\boldsymbol{\theta}}), \boldsymbol{I}^{-1}\nabla_{\boldsymbol{\theta}}\ell(\boldsymbol{z}_{\text{train}}^{(i)}; \widehat{\boldsymbol{\theta}}) \rangle, \tag{7}$$

where $\boldsymbol{I}$ is the Fisher information matrix of the loss $\mathcal{L}_{\text{train}}$.

**Grad-Dot, Grad-Cos (Perronnin et al., 2010; Yeh et al., 2018; Charpiat et al., 2019)**  Charpiat et al. (2019) proposed to measure relevance according to "how largely the loss will decrease when a small update is added to the model using the training instance." This can be computed as the dot product of the loss gradients, which we refer to as Grad-Dot.

$$R_{\text{GD}}(\boldsymbol{z}_{\text{test}}, \boldsymbol{z}_{\text{train}}) := \langle \nabla_{\boldsymbol{\theta}}\ell(\boldsymbol{z}_{\text{test}}; \widehat{\boldsymbol{\theta}}), \nabla_{\boldsymbol{\theta}}\ell(\boldsymbol{z}_{\text{train}}; \widehat{\boldsymbol{\theta}}) \rangle. \tag{8}$$

Note that a similar metric is studied by Yeh et al. (2018) as the *representer point value*.

As a modification of Grad-Dot, Charpiat et al. (2019) also proposed the following cosine version, which we refer to as Grad-Cos.

$$R_{\text{GC}}(\boldsymbol{z}_{\text{test}}, \boldsymbol{z}_{\text{train}}) := \cos(\nabla_{\boldsymbol{\theta}}\ell(\boldsymbol{z}_{\text{test}}; \widehat{\boldsymbol{\theta}}), \nabla_{\boldsymbol{\theta}}\ell(\boldsymbol{z}_{\text{train}}; \widehat{\boldsymbol{\theta}})). \tag{9}$$

Note that the use of the cosine between the gradients is also proposed by Perronnin et al. (2010).

## B EXPERIMENTAL SETUP

### B.1 DATASETS AND MODELS

**MNIST (LeCun et al., 1998)**  The MNIST dataset is used for handwritten digit image classification tasks. Here, input $\boldsymbol{x}$ is an image of a handwritten digit, and the output $y$ consists of 10 classes ("0"

---

[6]This metric is called $\ell$-RelatIF by Barshan et al. (2020)

to "9"). We adopted logistic regression and a CNN as the classification models. The CNN has six convolutional layers, and max-pooling layers for each two convolutional layers. The features obtained by these layers are fed into the global average pooling layer followed by a single linear layer. The number of the output channels of all the convolutional layers is set to 16. We trained the models using the Adam optimizer with a learning rate of 0.001. We used randomly sampled 5,500 training instances to train the models.

**CIFAR10 (Krizhevsky, 2009)**   The CIFAR10 dataset is used for object recognition tasks. Here, input $x$ is an image containing a certain object, and output $y$ consists of 10 classes, e.g., "bird" or "airplane." Note that we used the same models as for the MNIST dataset. In addition, we adopted MobileNetV2 (Sandler et al., 2018) as a model with a higher performance than the previous model. We trained the models using the Adam optimizer with a learning rate of 0.001. In the experiments, we first pre-trained the models using all the training instances of CIFAR10, and then trained the models using randomly sampled 5,000 training instances. Without the pre-training, the classification performance of the models dropped significantly.

Note that we did not examine IF and FK on MobileNetV2 because the matrix inverse in these metrics required too much time to calculate even with the conjugate gradient approximation proposed by Koh & Liang (2017).

**TREC (Li & Roth, 2002)**   The TREC dataset is used for question classification tasks. Here, input $x$ is a question sentence, and output $y$ is a question category consisting of six classes, e.g., "LOC" and "NUM." We used bag-of-words logistic regression and a two-layer Bi-LSTM as the classification models. In the Bi-LSTM, the last state is fed into one linear layer. The word embedding dimension is set to 16, and the dimension of the LSTM is set to 16 also. We trained the models using the Adam optimizer with a learning rate of 0.001. We used randomly sampled 2,726 training instances to train the models.

**AGNews (Zhang et al., 2015)**   The AGNews dataset is used for news article classification tasks. Here, input $x$ is a sentence, and output $y$ is a category comprising four classes, e.g., "business" and "sports." We used the same models as TREC. We trained the models using the Adam optimizer with a learning rate of 0.001. We used randomly sampled 6,000 training instances to train the models.

**Vehicle (Dua & Graff, 2017)**   The vehicle dataset is used for vehicle type classification tasks. Here, the input $x$ consists of 18 features, and the output $y$ is a type of vehicle comprising four classes, e.g., "bus" and "van." We used logistic regression and a three-layer MLP as the classification models. We trained the models using the Adam optimizer with a learning rate of 0.001. We used randomly sampled 423 training instances to train the models.

**Segment (Dua & Graff, 2017)**   The segment dataset is used for image classification tasks. Here, the input $x$ consists of 19 features, and the output $y$ consists of seven classes, e.g., "sky" and "window." We used the same models as Vehicle. We trained the models using the Adam optimizer with a learning rate of 0.001. We used randomly sampled 924 training instances to train the models.

### B.2   COMPUTING INFRASTRUCTURE

In our experiments, training of the models was run on a NVIDIA GTX 1080 GPU with Intel Xeon Silver 4112 CPU and 64GB RAM. Testing and computing relevance metrics were run on Xeon E5-2680 v2 CPU with 256GB RAM.

## C   DERIVATION OF GC FOR LINEAR LOGISTIC REGRESSION

We consider linear logistic regression whose conditional distribution $p(y \mid x; \theta)$ is given by the $y$-th entry of $\sigma(Wx)$, where $\sigma$ is the softmax function, $\theta = W \in \mathbb{R}^{C \times d}$, and $C$ and $d$ are the number of classes and the dimensionality of $x$, respectively. Recall that the cross entropy loss for linear logistic

regression is given as

$$\ell(\boldsymbol{z}; \boldsymbol{\theta}) = -\sum_{c=1}^{C} y_c \langle \boldsymbol{w}_c, \boldsymbol{x} \rangle + \log \sum_{c'=1}^{C} \exp(\langle \boldsymbol{w}_{c'}, \boldsymbol{x} \rangle), \tag{10}$$

where $W = [\boldsymbol{w}_1, \boldsymbol{w}_2, \dots, \boldsymbol{w}_C]^\top$. Let $\boldsymbol{e}_y$ be a vector whose $y$-th entry is one and zero otherwise. Then, the gradient of the loss with respect to $\boldsymbol{w}_c$ can be expressed as

$$\nabla_{\boldsymbol{w}_c} \ell(\boldsymbol{z}; \boldsymbol{\theta}) = (\sigma(W\boldsymbol{x}) - (\boldsymbol{e}_y)_c)\boldsymbol{x} = (\boldsymbol{r}^{\boldsymbol{z}})_c \boldsymbol{x}, \tag{11}$$

where $\boldsymbol{r}^{\boldsymbol{z}} = \sigma(W\boldsymbol{x}) - \boldsymbol{e}_y$ is the residual for the prediction on $\boldsymbol{z}$. Hence, we have

$$\langle \nabla_{\boldsymbol{\theta}} \ell(\boldsymbol{z}; \boldsymbol{\theta}), \nabla_{\boldsymbol{\theta}} \ell(\boldsymbol{z}'; \boldsymbol{\theta}) \rangle = \sum_{c=1}^{C} \langle \nabla_{\boldsymbol{w}_c} \ell(\boldsymbol{z}; \boldsymbol{\theta}), \nabla_{\boldsymbol{w}_c} \ell(\boldsymbol{z}'; \boldsymbol{\theta}) \rangle \tag{12}$$

$$= \sum_{c=1}^{C} (\boldsymbol{r}^{\boldsymbol{z}})_c (\boldsymbol{r}^{\boldsymbol{z}'})_c \langle \boldsymbol{x}, \boldsymbol{x}' \rangle \tag{13}$$

$$= \langle \boldsymbol{r}^{\boldsymbol{z}}, \boldsymbol{r}^{\boldsymbol{z}'} \rangle \langle \boldsymbol{x}, \boldsymbol{x}' \rangle, \tag{14}$$

which yields

$$R_{\mathrm{GC}}(\boldsymbol{z}, \boldsymbol{z}') = \frac{\langle \boldsymbol{r}^{\boldsymbol{z}}, \boldsymbol{r}^{\boldsymbol{z}'} \rangle \langle \boldsymbol{x}, \boldsymbol{x}' \rangle}{\|\boldsymbol{r}^{\boldsymbol{z}}\| \|\boldsymbol{x}\| \|\boldsymbol{r}^{\boldsymbol{z}'}\| \|\boldsymbol{x}'\|} \tag{15}$$

$$= \cos(\boldsymbol{r}^{\boldsymbol{z}}, \boldsymbol{r}^{\boldsymbol{z}'}) \cos(\boldsymbol{x}, \boldsymbol{x}'). \tag{16}$$

## D  REPAIRING GRADIENT-BASED METRICS

As described in Section 5, we found that training instances with extremely large norms were selected as relevant by IF, FK, and GD. Thus, to repair these metrics, we need to design metrics that can ignore instances with large norms. A simple yet effective way of repairing the metrics is to use $\ell_2$ or cosine instead of the dot product. As Figure 4 shows, the $\ell_2$ and cosine metrics performed better than the dot metrics. Indeed, the $\ell_2$ metrics do not favor instances with large norms that lead to large $\ell_2$-distance, and, through normalization, the cosine metrics completely ignore the effect of the norms

We name the repaired metrics of IF, FK, and GD based on the $\ell_2$ metric as $\ell_2^{\mathrm{IF}}$, $\ell_2^{\mathrm{FK}}$, and $\ell_2^{\mathrm{GD}}$, respectively, and the repaired metrics based on the cosine metric as $\cos^{\mathrm{IF}}$ and $\cos^{\mathrm{FK}}$, and $\cos^{\mathrm{GD}}$, respectively[7]. We observed that these repaired metrics attained higher success rates on several evaluation criteria. The details of the results can be found in Appendix F.

## E  DO THE MODELS CAPTURE SUBCLASSES?

The identical subclass test requires the model to obtain internal representations that can distinguish subclasses. Here, we confirm that this condition is satisfied for all the datasets and models we used in the experiments. We consider that the model captures the subclasses if the latent representation $\boldsymbol{h}^{\mathrm{all}}$ has cluster structures. Figure 9 visualizes $\boldsymbol{h}^{\mathrm{all}}$ for each dataset and model using UMAP (McInnes et al., 2018). The figures show that the instances from different subclasses are not mixed completely random. MNIST and TREC have relatively clear cluster structures, while CIFAR10 and AGNews have vague clusters without explicit boundaries. These figures imply that the models capture subclases (although it may not be perfect).

## F  COMPLETE EVALUATION RESULTS

### F.1  FULL RESULTS

We show the complete results of the model randomization test in Table 2, the identical class test in Table 3, and the identical subclass test in Table 4. The results we present here are consistent with our observations in Section 4.

---

[7]Note that $\cos^{\mathrm{IF}}$ is the same as RIF and $\cos^{\mathrm{GD}}$ is the same as GC.

Table 2: Average Spearman rank correlation coefficients $\pm$ std. of each similarity function for model randomization test. The metrics prefixed with $\Diamond$ are the ones we have repaired. The results with the average score in the 95% confidence interval of the null distribution that the correlation is zero, which is [-0.088, 0.088], are colored.

| | MNIST | | CIFAR10 | | | TREC | |
|---|---|---|---|---|---|---|---|
| Model | CNN | logreg | MobilenetV2 | CNN | logreg | Bi-LSTM | logreg |
| Parameter size | 12K | 8K | 2.2M | 12K | 31K | 20K | 7K |
| Accuracy | $0.98 \pm 0.00$ | $0.92 \pm 0.00$ | $0.89 \pm 0.01$ | $0.72 \pm 0.02$ | $0.35 \pm 0.01$ | $0.86 \pm 0.01$ | $0.81 \pm 0.02$ |
| $\ell_2^x$ | $1.00 \pm .00$ | $1.00 \pm .00$ | $1.00 \pm .00$ | $1.00 \pm .00$ | $1.00 \pm .00$ | $1.00 \pm .00$ | $1.00 \pm .00$ |
| $\ell_2^{\text{last}}$ | $.15 \pm .01$ | - | $.07 \pm .00$ | $.05 \pm .01$ | - | $.19 \pm .01$ | - |
| $\ell_2^{\text{all}}$ | $.79 \pm .00$ | - | $.02 \pm .01$ | $.13 \pm .01$ | - | $.25 \pm .02$ | - |
| $\cos^x$ | $1.00 \pm .00$ | $1.00 \pm .00$ | $1.00 \pm .00$ | $1.00 \pm .00$ | $1.00 \pm .00$ | $1.00 \pm .00$ | $1.00 \pm .00$ |
| $\cos^{\text{last}}$ | $.24 \pm .00$ | - | $.07 \pm .01$ | $.04 \pm .01$ | - | $.17 \pm .02$ | - |
| $\cos^{\text{all}}$ | $.78 \pm .00$ | - | $.02 \pm .01$ | $.09 \pm .01$ | - | $.26 \pm .03$ | - |
| $\text{dot}^x$ | $1.00 \pm .00$ | $1.00 \pm .00$ | $1.00 \pm .00$ | $1.00 \pm .00$ | $1.00 \pm .00$ | $1.00 \pm .00$ | $1.00 \pm .00$ |
| $\text{dot}^{\text{last}}$ | $.39 \pm .01$ | - | $.05 \pm .01$ | $.04 \pm .01$ | - | $.25 \pm .01$ | - |
| $\text{dot}^{\text{all}}$ | $.80 \pm .00$ | - | $-.00 \pm .01$ | $.12 \pm .01$ | - | $.26 \pm .03$ | - |
| IF | $.05 \pm .00$ | $-.00 \pm .00$ | $-.05 \pm .01$ | $-.04 \pm .01$ | $-.04 \pm .01$ | $.01 \pm .01$ | $.06 \pm .01$ |
| $\Diamond \ell_2^{\text{IF}}$ | $.00 \pm .02$ | $-.11 \pm .00$ | $.01 \pm .02$ | $-.03 \pm .02$ | $-.05 \pm .01$ | $-.00 \pm .02$ | $-.13 \pm .02$ |
| $\Diamond \cos^{\text{IF}}$ | $.02 \pm .00$ | $-.05 \pm .00$ | $.04 \pm .01$ | $.03 \pm .01$ | $-.03 \pm .01$ | $-.01 \pm .01$ | $.03 \pm .01$ |
| FK | $-.02 \pm .01$ | $.02 \pm .01$ | $-.02 \pm .01$ | $.01 \pm .01$ | $-.03 \pm .01$ | $.01 \pm .00$ | $.03 \pm .00$ |
| $\Diamond \ell_2^{\text{FK}}$ | $-.10 \pm .04$ | $.05 \pm .00$ | $-.16 \pm .05$ | $-.12 \pm .02$ | $.03 \pm .01$ | $-.14 \pm .03$ | $.15 \pm .01$ |
| $\Diamond \cos^{\text{FK}}$ | $-.00 \pm .02$ | $.05 \pm .01$ | $-.05 \pm .01$ | $-.03 \pm .01$ | $-.01 \pm .01$ | $-.07 \pm .02$ | $-.03 \pm .00$ |
| GD | $-.08 \pm .02$ | $.01 \pm .01$ | $-.03 \pm .01$ | $-.02 \pm .01$ | $.04 \pm .01$ | $-.04 \pm .01$ | $-.02 \pm .02$ |
| GC | $-.07 \pm .03$ | $-.03 \pm .01$ | $-.02 \pm .02$ | $.01 \pm .03$ | $-.05 \pm .01$ | $-.04 \pm .02$ | $-.01 \pm .01$ |
| $\Diamond \ell_2^{\text{grad}}$ | $-.09 \pm .04$ | $-.13 \pm .01$ | $-.09 \pm .04$ | $-.07 \pm .02$ | $-.06 \pm .01$ | $-.02 \pm .02$ | $-.10 \pm .02$ |

| | AGNews | | Vehicle | | Segment | |
|---|---|---|---|---|---|---|
| Model | Bi-LSTM | logreg | MLP | logreg | MLP | logreg |
| Parameter size | 27K | 9K | 1K | 76 | 1K | 140 |
| Accuracy | $0.80 \pm 0.02$ | $0.80 \pm 0.01$ | $0.77 \pm 0.02$ | $0.77 \pm 0.01$ | $0.98 \pm 0.01$ | $0.97 \pm 0.00$ |
| $\ell_2^x$ | $1.00 \pm .00$ | $1.00 \pm .00$ | $1.00 \pm .00$ | $1.00 \pm .00$ | $1.00 \pm .00$ | $1.00 \pm .00$ |
| $\ell_2^{\text{last}}$ | $.07 \pm .01$ | - | $.16 \pm .04$ | - | $.62 \pm .15$ | - |
| $\ell_2^{\text{all}}$ | $.17 \pm .01$ | - | $.20 \pm .10$ | - | $.78 \pm .08$ | - |
| $\cos^x$ | $1.00 \pm .00$ | $1.00 \pm .00$ | $1.00 \pm .00$ | $1.00 \pm .00$ | $1.00 \pm .00$ | $1.00 \pm .00$ |
| $\cos^{\text{last}}$ | $.07 \pm .01$ | - | $-.09 \pm .18$ | - | $.60 \pm .09$ | - |
| $\cos^{\text{all}}$ | $.12 \pm .02$ | - | $-.01 \pm .13$ | - | $.77 \pm .06$ | - |
| $\text{dot}^x$ | $1.00 \pm .00$ | $1.00 \pm .00$ | $1.00 \pm .00$ | $1.00 \pm .00$ | $1.00 \pm .00$ | $1.00 \pm .00$ |
| $\text{dot}^{\text{last}}$ | $.07 \pm .01$ | - | $.85 \pm .33$ | - | $.61 \pm .23$ | - |
| $\text{dot}^{\text{all}}$ | $.20 \pm .01$ | - | $.97 \pm .03$ | - | $.72 \pm .16$ | - |
| IF | $.03 \pm .01$ | $.05 \pm .01$ | $-.01 \pm .03$ | $-.01 \pm .02$ | $.00 \pm .01$ | $.01 \pm .02$ |
| $\Diamond \ell_2^{\text{IF}}$ | $-.04 \pm .02$ | $-.13 \pm .00$ | $-.18 \pm .24$ | $-.01 \pm .28$ | $.03 \pm .13$ | $-.10 \pm .26$ |
| $\Diamond \cos^{\text{IF}}$ | $.02 \pm .01$ | $.03 \pm .01$ | $-.01 \pm .03$ | $-.01 \pm .05$ | $.04 \pm .10$ | $.01 \pm .05$ |
| FK | $.04 \pm .01$ | $.03 \pm .00$ | $.01 \pm .06$ | $.02 \pm .07$ | $-.01 \pm .02$ | $-.00 \pm .02$ |
| $\Diamond \ell_2^{\text{FK}}$ | $-.17 \pm .04$ | $.14 \pm .00$ | $-.18 \pm .21$ | $-.01 \pm .17$ | $.05 \pm .07$ | $-.02 \pm .20$ |
| $\Diamond \cos^{\text{FK}}$ | $-.00 \pm .03$ | $-.03 \pm .00$ | $.08 \pm .13$ | $-.04 \pm .12$ | $-.01 \pm .03$ | $.01 \pm .04$ |
| GD | $-.04 \pm .01$ | $.03 \pm .01$ | $.01 \pm .11$ | $-.02 \pm .05$ | $.01 \pm .03$ | $.02 \pm .03$ |
| GC | $.01 \pm .02$ | $.04 \pm .01$ | $.02 \pm .13$ | $-.06 \pm .11$ | $.00 \pm .06$ | $.01 \pm .05$ |
| $\Diamond \ell_2^{\text{grad}}$ | $-.01 \pm .02$ | $-.14 \pm .00$ | $-.13 \pm .21$ | $.11 \pm .23$ | $.02 \pm .12$ | $-.09 \pm .21$ |

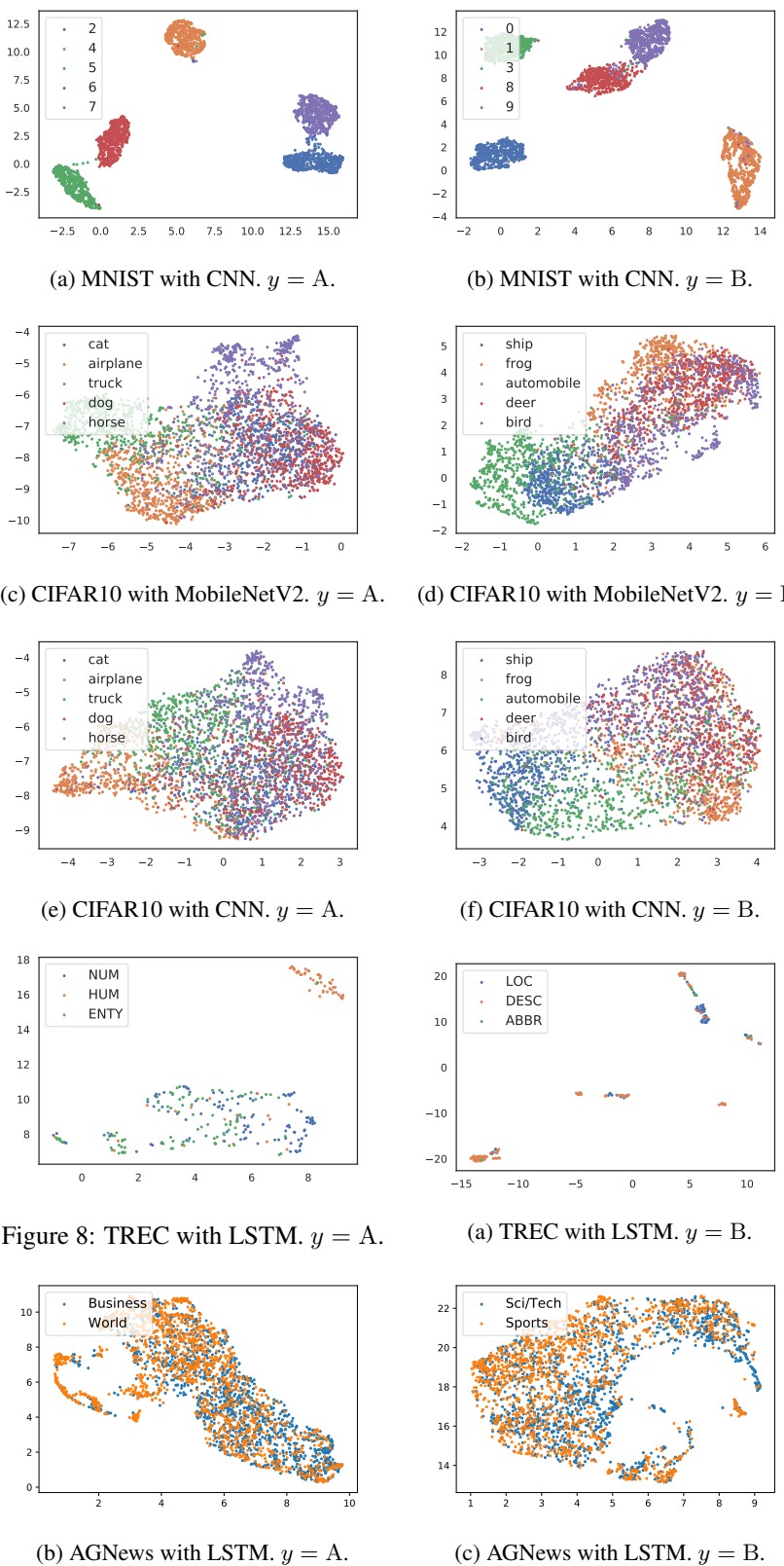

(a) MNIST with CNN. $y = A$.

(b) MNIST with CNN. $y = B$.

(c) CIFAR10 with MobileNetV2. $y = A$.

(d) CIFAR10 with MobileNetV2. $y = B$.

(e) CIFAR10 with CNN. $y = A$.

(f) CIFAR10 with CNN. $y = B$.

Figure 8: TREC with LSTM. $y = A$.

(a) TREC with LSTM. $y = B$.

(b) AGNews with LSTM. $y = A$.

(c) AGNews with LSTM. $y = B$.

Figure 9: visualization of $\boldsymbol{h}^{\mathrm{all}}$ in each dataset and model using UMAP.

Table 3: Average success rate $\pm$ std. of each relevancy metric for identical class test. The metrics prefixed with $\diamondsuit$ are the ones we have repaired. The results with the average success rate over 0.5 are colored.

| | MNIST | | CIFAR10 | | | TREC | |
|---|---|---|---|---|---|---|---|
| Model | CNN | logreg | MobilenetV2 | CNN | logreg | Bi-LSTM | logreg |
| Parameter size | 12K | 8K | 2.2M | 12K | 31K | 20K | 7K |
| Accuracy | $0.98 \pm 0.00$ | $0.92 \pm 0.00$ | $0.89 \pm 0.01$ | $0.72 \pm 0.02$ | $0.35 \pm 0.01$ | $0.86 \pm 0.01$ | $0.81 \pm 0.02$ |
| $\ell_2^x$ | $.93 \pm .01$ | $.88 \pm .01$ | $.26 \pm .02$ | $.26 \pm .02$ | $.24 \pm .02$ | $.70 \pm .00$ | $.75 \pm .00$ |
| $\ell_2^{last}$ | $.99 \pm .01$ | - | $1.00 \pm .00$ | $.75 \pm .02$ | - | $.89 \pm .00$ | - |
| $\ell_2^{all}$ | $.98 \pm .00$ | - | $.93 \pm .02$ | $.61 \pm .02$ | - | $.88 \pm .00$ | - |
| $\cos^x$ | $.94 \pm .01$ | $.88 \pm .01$ | $.30 \pm .03$ | $.29 \pm .02$ | $.26 \pm .02$ | $.73 \pm .00$ | $.76 \pm .00$ |
| $\cos^{last}$ | $.99 \pm .01$ | - | $1.00 \pm .00$ | $.78 \pm .02$ | - | $.89 \pm .00$ | - |
| $\cos^{all}$ | $.98 \pm .00$ | - | $.97 \pm .01$ | $.71 \pm .02$ | - | $.90 \pm .00$ | - |
| $\mathrm{dot}^x$ | $.69 \pm .02$ | $.68 \pm .02$ | $.09 \pm .02$ | $.10 \pm .01$ | $.11 \pm .02$ | $.33 \pm .00$ | $.34 \pm .00$ |
| $\mathrm{dot}^{last}$ | $.67 \pm .02$ | - | $1.00 \pm .00$ | $.20 \pm .02$ | - | $.93 \pm .00$ | - |
| $\mathrm{dot}^{all}$ | $.96 \pm .01$ | - | $.96 \pm .01$ | $.31 \pm .01$ | - | $.93 \pm .00$ | - |
| IF | $.09 \pm .01$ | $.26 \pm .02$ | - | $.10 \pm .01$ | $.09 \pm .02$ | $.29 \pm .00$ | $.86 \pm .00$ |
| $\diamondsuit\, \ell_2^{IF}$ | $.72 \pm .01$ | $.62 \pm .02$ | - | $.14 \pm .01$ | $.14 \pm .01$ | $.98 \pm .00$ | $.95 \pm .00$ |
| $\diamondsuit\, \cos^{IF}$ | $.82 \pm .01$ | $.69 \pm .02$ | - | $.12 \pm .01$ | $.13 \pm .02$ | $.99 \pm .00$ | $.96 \pm .00$ |
| FK | $.10 \pm .01$ | $.21 \pm .02$ | - | $.20 \pm .02$ | $.20 \pm .02$ | $.28 \pm .00$ | $.24 \pm .00$ |
| $\diamondsuit\, \ell_2^{FK}$ | $.77 \pm .02$ | $.93 \pm .01$ | - | $.82 \pm .01$ | $.98 \pm .00$ | $.99 \pm .00$ | $.96 \pm .00$ |
| $\diamondsuit\, \cos^{FK}$ | $.92 \pm .01$ | $.97 \pm .01$ | - | $.93 \pm .01$ | $.99 \pm .00$ | $1.00 \pm .00$ | $.96 \pm .00$ |
| GD | $.30 \pm .01$ | $.87 \pm .01$ | $.26 \pm .03$ | $.71 \pm .02$ | $1.00 \pm .00$ | $.49 \pm .00$ | $1.00 \pm .00$ |
| GC | $.99 \pm .00$ | $1.00 \pm .00$ | $.99 \pm .00$ | $.99 \pm .00$ | $1.00 \pm .00$ | $1.00 \pm .00$ | $1.00 \pm .00$ |
| $\diamondsuit\, \ell_2^{grad}$ | $.94 \pm .01$ | $.99 \pm .00$ | $.97 \pm .01$ | $.99 \pm .00$ | $1.00 \pm .00$ | $1.00 \pm .00$ | $1.00 \pm .00$ |

| | AGNews | | Vehicle | | Segment | |
|---|---|---|---|---|---|---|
| Model | Bi-LSTM | logreg | MLP | logreg | MLP | logreg |
| Parameter size | 27K | 9K | 1K | 76 | 1K | 140 |
| Accuracy | $0.80 \pm 0.02$ | $0.80 \pm 0.01$ | $0.77 \pm 0.02$ | $0.77 \pm 0.01$ | $0.98 \pm 0.01$ | $0.97 \pm 0.00$ |
| $\ell_2^x$ | $.39 \pm .02$ | $.40 \pm .02$ | $.65 \pm .02$ | $.62 \pm .02$ | $.93 \pm .01$ | $.92 \pm .01$ |
| $\ell_2^{last}$ | $.84 \pm .02$ | - | $.72 \pm .03$ | - | $.97 \pm .01$ | - |
| $\ell_2^{all}$ | $.84 \pm .01$ | - | $.72 \pm .03$ | - | $.96 \pm .01$ | - |
| $\cos^x$ | $.47 \pm .01$ | $.51 \pm .02$ | $.66 \pm .02$ | $.63 \pm .02$ | $.91 \pm .01$ | $.90 \pm .01$ |
| $\cos^{last}$ | $.85 \pm .01$ | - | $.74 \pm .04$ | - | $.97 \pm .01$ | - |
| $\cos^{all}$ | $.84 \pm .01$ | - | $.73 \pm .04$ | - | $.96 \pm .01$ | - |
| $\mathrm{dot}^x$ | $.28 \pm .02$ | $.47 \pm .02$ | $.25 \pm .00$ | $.27 \pm .01$ | $.37 \pm .01$ | $.37 \pm .01$ |
| $\mathrm{dot}^{last}$ | $.89 \pm .01$ | - | $.26 \pm .02$ | - | $.17 \pm .06$ | - |
| $\mathrm{dot}^{all}$ | $.90 \pm .01$ | - | $.27 \pm .06$ | - | $.13 \pm .01$ | - |
| IF | $.24 \pm .01$ | $.67 \pm .02$ | $.39 \pm .16$ | $.78 \pm .08$ | $.15 \pm .03$ | $.52 \pm .07$ |
| $\diamondsuit\, \ell_2^{IF}$ | $.99 \pm .00$ | $.92 \pm .01$ | $.88 \pm .06$ | $.95 \pm .01$ | $.79 \pm .13$ | $.80 \pm .05$ |
| $\diamondsuit\, \cos^{IF}$ | $1.00 \pm .00$ | $.97 \pm .01$ | $.96 \pm .02$ | $.99 \pm .01$ | $.84 \pm .11$ | $.92 \pm .08$ |
| FK | $.32 \pm .01$ | $.29 \pm .03$ | $.31 \pm .18$ | $.26 \pm .17$ | $.15 \pm .04$ | $.17 \pm .10$ |
| $\diamondsuit\, \ell_2^{FK}$ | $.94 \pm .01$ | $.68 \pm .02$ | $.93 \pm .04$ | $.94 \pm .03$ | $.86 \pm .06$ | $.95 \pm .02$ |
| $\diamondsuit\, \cos^{FK}$ | $.95 \pm .01$ | $.84 \pm .02$ | $.99 \pm .01$ | $.99 \pm .01$ | $.97 \pm .02$ | $.99 \pm .01$ |
| GD | $.76 \pm .01$ | $1.00 \pm .00$ | $.90 \pm .10$ | $.98 \pm .02$ | $.30 \pm .14$ | $.55 \pm .11$ |
| GC | $1.00 \pm .00$ | $1.00 \pm .00$ | $1.00 \pm .00$ | $1.00 \pm .00$ | $.97 \pm .02$ | $1.00 \pm .00$ |
| $\diamondsuit\, \ell_2^{grad}$ | $1.00 \pm .00$ | $1.00 \pm .00$ | $.99 \pm .01$ | $1.00 \pm .00$ | $.90 \pm .05$ | $.99 \pm .01$ |

Table 4: Average success rate $\pm$ std. of each relevancy metric for identical subclass test. The metrics prefixed with $\diamondsuit$ are the ones we have repaired. The results with the average success rate over 0.5 are colored.

| | MNIST | | CIFAR10 | | | TREC | |
|---|---|---|---|---|---|---|---|
| Model | CNN | logreg | MobilenetV2 | CNN | logreg | Bi-LSTM | logreg |
| Parameter size | 12K | 8K | 2.2M | 12K | 31K | 20K | 7K |
| Accuracy | $0.99 \pm 0.00$ | $0.88 \pm 0.01$ | $0.92 \pm 0.01$ | $0.84 \pm 0.03$ | $0.71 \pm 0.03$ | $0.86 \pm 0.01$ | $0.81 \pm 0.02$ |
| $\ell_2^x$ | $.93 \pm .01$ | $.96 \pm .02$ | $.26 \pm .02$ | $.29 \pm .04$ | $.31 \pm .03$ | $.78 \pm .03$ | $.78 \pm .02$ |
| $\ell_2^{\text{last}}$ | $.89 \pm .02$ | - | $.29 \pm .04$ | $.35 \pm .04$ | - | $.76 \pm .02$ | - |
| $\ell_2^{\text{all}}$ | $.97 \pm .01$ | - | $.49 \pm .04$ | $.38 \pm .03$ | - | $.77 \pm .03$ | - |
| $\cos^x$ | $.95 \pm .01$ | $.96 \pm .02$ | $.29 \pm .03$ | $.31 \pm .04$ | $.31 \pm .03$ | $.82 \pm .02$ | $.81 \pm .02$ |
| $\cos^{\text{last}}$ | $.89 \pm .02$ | - | $.32 \pm .03$ | $.33 \pm .03$ | - | $.75 \pm .02$ | - |
| $\cos^{\text{all}}$ | $.98 \pm .00$ | - | $.71 \pm .04$ | $.50 \pm .03$ | - | $.77 \pm .02$ | - |
| $\text{dot}^x$ | $.70 \pm .03$ | $.75 \pm .03$ | $.09 \pm .02$ | $.11 \pm .03$ | $.09 \pm .02$ | $.33 \pm .03$ | $.34 \pm .03$ |
| $\text{dot}^{\text{last}}$ | $.24 \pm .04$ | - | $.22 \pm .02$ | $.20 \pm .01$ | - | $.40 \pm .03$ | - |
| $\text{dot}^{\text{all}}$ | $.94 \pm .01$ | - | $.68 \pm .03$ | $.25 \pm .03$ | - | $.59 \pm .03$ | - |
| IF | $.12 \pm .01$ | $.39 \pm .05$ | - | $.06 \pm .02$ | $.08 \pm .02$ | $.31 \pm .02$ | $.49 \pm .03$ |
| $\diamondsuit\, \ell_2^{\text{IF}}$ | $.62 \pm .04$ | $.76 \pm .03$ | - | $.17 \pm .02$ | $.12 \pm .02$ | $.68 \pm .02$ | $.79 \pm .02$ |
| $\diamondsuit\, \cos^{\text{IF}}$ | $.70 \pm .02$ | $.87 \pm .02$ | - | $.15 \pm .02$ | $.09 \pm .02$ | $.72 \pm .01$ | $.75 \pm .03$ |
| FK | $.19 \pm .03$ | $.14 \pm .02$ | - | $.11 \pm .01$ | $.11 \pm .02$ | $.30 \pm .02$ | $.16 \pm .02$ |
| $\diamondsuit\, \ell_2^{\text{FK}}$ | $.81 \pm .02$ | $.76 \pm .03$ | - | $.31 \pm .03$ | $.24 \pm .02$ | $.73 \pm .03$ | $.78 \pm .02$ |
| $\diamondsuit\, \cos^{\text{FK}}$ | $.91 \pm .02$ | $.85 \pm .02$ | - | $.37 \pm .03$ | $.23 \pm .02$ | $.81 \pm .02$ | $.79 \pm .01$ |
| GD | $.42 \pm .05$ | $.48 \pm .03$ | $.20 \pm .02$ | $.24 \pm .03$ | $.21 \pm .04$ | $.45 \pm .02$ | $.60 \pm .02$ |
| GC | $.97 \pm .01$ | $.98 \pm .01$ | $.54 \pm .03$ | $.43 \pm .04$ | $.39 \pm .03$ | $.81 \pm .01$ | $.87 \pm .02$ |
| $\diamondsuit\, \ell_2^{\text{grad}}$ | $.91 \pm .02$ | $.95 \pm .01$ | $.28 \pm .03$ | $.38 \pm .03$ | $.34 \pm .03$ | $.78 \pm .02$ | $.88 \pm .02$ |

| | AGNews | | Vehicle | | Segment | |
|---|---|---|---|---|---|---|
| Model | Bi-LSTM | logreg | MLP | logreg | MLP | logreg |
| Parameter size | 27K | 9K | 1K | 38 | 1K | 40 |
| Accuracy | $0.80 \pm 0.02$ | $0.80 \pm 0.01$ | $0.73 \pm 0.02$ | $0.73 \pm 0.01$ | $0.94 \pm 0.01$ | $0.90 \pm 0.01$ |
| $\ell_2^x$ | $.40 \pm .02$ | $.41 \pm .01$ | $.67 \pm .03$ | $.65 \pm .02$ | $.95 \pm .01$ | $.95 \pm .01$ |
| $\ell_2^{\text{last}}$ | $.53 \pm .02$ | - | $.64 \pm .05$ | - | $.95 \pm .02$ | - |
| $\ell_2^{\text{all}}$ | $.58 \pm .01$ | - | $.66 \pm .04$ | - | $.96 \pm .01$ | - |
| $\cos^x$ | $.49 \pm .02$ | $.53 \pm .02$ | $.68 \pm .04$ | $.66 \pm .03$ | $.92 \pm .01$ | $.93 \pm .01$ |
| $\cos^{\text{last}}$ | $.54 \pm .01$ | - | $.68 \pm .06$ | - | $.93 \pm .01$ | - |
| $\cos^{\text{all}}$ | $.59 \pm .02$ | - | $.67 \pm .03$ | - | $.94 \pm .01$ | - |
| $\text{dot}^x$ | $.28 \pm .02$ | $.48 \pm .02$ | $.26 \pm .03$ | $.26 \pm .03$ | $.38 \pm .01$ | $.41 \pm .02$ |
| $\text{dot}^{\text{last}}$ | $.52 \pm .02$ | - | $.27 \pm .03$ | - | $.15 \pm .03$ | - |
| $\text{dot}^{\text{all}}$ | $.54 \pm .02$ | - | $.28 \pm .04$ | - | $.13 \pm .02$ | - |
| IF | $.25 \pm .02$ | $.48 \pm .01$ | $.34 \pm .12$ | $.54 \pm .09$ | $.16 \pm .02$ | $.49 \pm .08$ |
| $\diamondsuit\, \ell_2^{\text{IF}}$ | $.56 \pm .02$ | $.77 \pm .02$ | $.76 \pm .14$ | $.86 \pm .06$ | $.65 \pm .10$ | $.43 \pm .12$ |
| $\diamondsuit\, \cos^{\text{IF}}$ | $.56 \pm .02$ | $.80 \pm .02$ | $.86 \pm .09$ | $.91 \pm .08$ | $.62 \pm .11$ | $.86 \pm .05$ |
| FK | $.28 \pm .01$ | $.25 \pm .02$ | $.16 \pm .08$ | $.20 \pm .05$ | $.16 \pm .07$ | $.10 \pm .05$ |
| $\diamondsuit\, \ell_2^{\text{FK}}$ | $.56 \pm .02$ | $.63 \pm .02$ | $.73 \pm .13$ | $.62 \pm .09$ | $.73 \pm .13$ | $.93 \pm .03$ |
| $\diamondsuit\, \cos^{\text{FK}}$ | $.61 \pm .02$ | $.73 \pm .02$ | $.80 \pm .06$ | $.67 \pm .10$ | $.81 \pm .10$ | $.96 \pm .02$ |
| GD | $.50 \pm .02$ | $.54 \pm .02$ | $.47 \pm .09$ | $.43 \pm .03$ | $.34 \pm .08$ | $.37 \pm .08$ |
| GC | $.65 \pm .02$ | $.72 \pm .02$ | $.82 \pm .06$ | $.83 \pm .07$ | $.81 \pm .10$ | $.96 \pm .01$ |
| $\diamondsuit\, \ell_2^{\text{grad}}$ | $.61 \pm .02$ | $.73 \pm .03$ | $.72 \pm .10$ | $.75 \pm .09$ | $.75 \pm .10$ | $.90 \pm .03$ |

## F.2   Additional Results

The identical class test require the most relevant instance to be of the same class as the test instance. In practice, users can be more confident about a model's output if several instances are provided as evidence. In other words, we expect that the most relevant and a first few relevant instances will be of the same class. This observation leads to the additional criterion, which is a generalization of the identical class test.

**Definition 5** (Top-$k$ Identical Class Test). For $z_{\text{test}} = (x_{\text{test}}, \widehat{y}_{\text{test}})$, let $\bar{z}^j = (\bar{x}^j, \bar{y}^j)$ be a training instance with the $j$-th largest relevance score. Then, we require $\bar{y}^j = \widehat{y}_{\text{test}}$ for any $j \in \{1, 2, \ldots, k\}$.

This observation also applies to identical subclass test, which leads to the following criterion

**Definition 6** (Top-$k$ Identical Subclass Test). For $z_{\text{test}} = (x_{\text{test}}, \widehat{y}_{\text{test}})$, let $\bar{z}^j = (\bar{x}^j, \bar{y}^j)$ be a training instance with the $j$-th largest relevance score. Then, we require $s(\bar{z}^j) = s(\widehat{z}_{\text{test}})$, $\forall j \in \{1, 2, \ldots, k\}$.

We show the results of the top-10 identical class test in Table 3, and the top-10 identical subclass test in Table 4.

## G   Examples of Each Explanation Method

We show some examples of the relevant instances using several relevance metrics on CIFAR10 with CNN in Figure 10 and Figure 11 and on AGNews with LSTM in Table 7 and Table 8. We show examples of both correct (in Figure 10 and Table 7) and incorrect (in Figure 11 and Table 8) predictions. As mentioned in Section 5, the relevance metrics based on the dot product of the gradient, such as IF, FK, and GD, tend to select instances with large norms, and therefore we can see that non-typical instances have been selected.

Table 5: Average success rate $\pm$ std. of each relevancy metric for top-10 identical class test. The metrics prefixed with $\Diamond$ are the ones we have repaired. The results with the average success rate over 0.5 are colored.

| | MNIST | | CIFAR10 | | | TREC | |
|---|---|---|---|---|---|---|---|
| Model | CNN | logreg | MobilenetV2 | CNN | logreg | Bi-LSTM | logreg |
| Parameter size | 12K | 8K | 2.2M | 12K | 31K | 20K | 7K |
| Accuracy | $0.98 \pm 0.00$ | $0.92 \pm 0.00$ | $0.89 \pm 0.01$ | $0.72 \pm 0.02$ | $0.35 \pm 0.01$ | $0.86 \pm 0.01$ | $0.81 \pm 0.02$ |
| $\ell_2^x$ | $.63 \pm .02$ | $.63 \pm .02$ | $.00 \pm .00$ | $.00 \pm .00$ | $.00 \pm .00$ | $.23 \pm .00$ | $.23 \pm .00$ |
| $\ell_2^{\text{last}}$ | $.95 \pm .01$ | - | $.98 \pm .01$ | $.30 \pm .01$ | - | $.68 \pm .00$ | - |
| $\ell_2^{\text{all}}$ | $.89 \pm .01$ | - | $.64 \pm .05$ | $.14 \pm .01$ | - | $.66 \pm .00$ | - |
| $\cos^x$ | $.67 \pm .02$ | $.65 \pm .02$ | $.00 \pm .00$ | $.00 \pm .00$ | $.00 \pm .00$ | $.24 \pm .00$ | $.24 \pm .00$ |
| $\cos^{\text{last}}$ | $.97 \pm .01$ | - | $.98 \pm .01$ | $.33 \pm .02$ | - | $.69 \pm .00$ | - |
| $\cos^{\text{all}}$ | $.92 \pm .00$ | - | $.84 \pm .03$ | $.23 \pm .02$ | - | $.68 \pm .00$ | - |
| $dot^x$ | $.19 \pm .01$ | $.20 \pm .02$ | $.00 \pm .00$ | $.00 \pm .00$ | $.00 \pm .00$ | $.05 \pm .00$ | $.05 \pm .00$ |
| $dot^{\text{last}}$ | $.42 \pm .03$ | - | $.98 \pm .01$ | $.04 \pm .01$ | - | $.75 \pm .00$ | - |
| $dot^{\text{all}}$ | $.88 \pm .01$ | - | $.79 \pm .03$ | $.05 \pm .01$ | - | $.84 \pm .00$ | - |
| IF | $.00 \pm .00$ | $.00 \pm .00$ | - | $.00 \pm .00$ | $.00 \pm .00$ | $.00 \pm .00$ | $.24 \pm .00$ |
| $\Diamond\,\ell_2^{\text{IF}}$ | $.25 \pm .01$ | $.10 \pm .01$ | - | $.00 \pm .00$ | $.00 \pm .00$ | $.83 \pm .00$ | $.47 \pm .00$ |
| $\Diamond\,\cos^{\text{IF}}$ | $.59 \pm .02$ | $.17 \pm .01$ | - | $.00 \pm .00$ | $.00 \pm .00$ | $.91 \pm .00$ | $.65 \pm .00$ |
| FK | $.00 \pm .00$ | $.02 \pm .01$ | - | $.00 \pm .00$ | $.06 \pm .01$ | $.01 \pm .00$ | $.00 \pm .00$ |
| $\Diamond\,\ell_2^{\text{FK}}$ | $.23 \pm .03$ | $.65 \pm .02$ | - | $.25 \pm .02$ | $.87 \pm .01$ | $.90 \pm .00$ | $.71 \pm .00$ |
| $\Diamond\,\cos^{\text{FK}}$ | $.59 \pm .01$ | $.82 \pm .02$ | - | $.54 \pm .02$ | $.93 \pm .01$ | $.95 \pm .00$ | $.77 \pm .00$ |
| GD | $.00 \pm .00$ | $.41 \pm .02$ | $.00 \pm .00$ | $.15 \pm .01$ | $1.00 \pm .00$ | $.11 \pm .00$ | $.99 \pm .00$ |
| GC | $.95 \pm .01$ | $.99 \pm .01$ | $.92 \pm .02$ | $.92 \pm .01$ | $1.00 \pm .00$ | $.96 \pm .00$ | $1.00 \pm .00$ |
| $\Diamond\,\ell_2^{\text{grad}}$ | $.57 \pm .02$ | $.95 \pm .01$ | $.78 \pm .03$ | $.80 \pm .01$ | $.99 \pm .00$ | $.94 \pm .00$ | $1.00 \pm .00$ |

| | AGNews | | Vehicle | | Segment | |
|---|---|---|---|---|---|---|
| Model | Bi-LSTM | logreg | MLP | logreg | MLP | logreg |
| Parameter size | 27K | 9K | 1K | 76 | 1K | 140 |
| Accuracy | $0.80 \pm 0.02$ | $0.80 \pm 0.01$ | $0.77 \pm 0.02$ | $0.77 \pm 0.01$ | $0.98 \pm 0.01$ | $0.97 \pm 0.00$ |
| $\ell_2^x$ | $.00 \pm .00$ | $.00 \pm .00$ | $.09 \pm .02$ | $.09 \pm .02$ | $.60 \pm .01$ | $.60 \pm .01$ |
| $\ell_2^{\text{last}}$ | $.48 \pm .03$ | - | $.19 \pm .07$ | - | $.77 \pm .03$ | - |
| $\ell_2^{\text{all}}$ | $.46 \pm .01$ | - | $.16 \pm .06$ | - | $.74 \pm .03$ | - |
| $\cos^x$ | $.01 \pm .00$ | $.02 \pm .01$ | $.10 \pm .02$ | $.10 \pm .01$ | $.44 \pm .02$ | $.44 \pm .02$ |
| $\cos^{\text{last}}$ | $.51 \pm .03$ | - | $.22 \pm .07$ | - | $.78 \pm .03$ | - |
| $\cos^{\text{all}}$ | $.48 \pm .02$ | - | $.17 \pm .06$ | - | $.72 \pm .04$ | - |
| $dot^x$ | $.01 \pm .00$ | $.01 \pm .00$ | $.15 \pm .12$ | $.16 \pm .13$ | $.23 \pm .02$ | $.23 \pm .02$ |
| $dot^{\text{last}}$ | $.64 \pm .03$ | - | $.13 \pm .11$ | - | $.05 \pm .06$ | - |
| $dot^{\text{all}}$ | $.66 \pm .03$ | - | $.15 \pm .11$ | - | $.00 \pm .01$ | - |
| IF | $.00 \pm .00$ | $.02 \pm .01$ | $.01 \pm .01$ | $.10 \pm .03$ | $.00 \pm .00$ | $.10 \pm .03$ |
| $\Diamond\,\ell_2^{\text{IF}}$ | $.94 \pm .01$ | $.20 \pm .02$ | $.25 \pm .13$ | $.47 \pm .05$ | $.32 \pm .15$ | $.48 \pm .05$ |
| $\Diamond\,\cos^{\text{IF}}$ | $.97 \pm .01$ | $.48 \pm .01$ | $.42 \pm .12$ | $.61 \pm .03$ | $.63 \pm .16$ | $.83 \pm .12$ |
| FK | $.00 \pm .00$ | $.00 \pm .00$ | $.05 \pm .11$ | $.08 \pm .11$ | $.00 \pm .01$ | $.03 \pm .06$ |
| $\Diamond\,\ell_2^{\text{FK}}$ | $.61 \pm .02$ | $.06 \pm .01$ | $.55 \pm .19$ | $.64 \pm .12$ | $.32 \pm .17$ | $.60 \pm .14$ |
| $\Diamond\,\cos^{\text{FK}}$ | $.71 \pm .03$ | $.15 \pm .01$ | $.85 \pm .06$ | $.85 \pm .08$ | $.78 \pm .08$ | $.92 \pm .03$ |
| GD | $.55 \pm .02$ | $.98 \pm .01$ | $.56 \pm .19$ | $.70 \pm .05$ | $.09 \pm .08$ | $.37 \pm .05$ |
| GC | $1.00 \pm .00$ | $1.00 \pm .00$ | $.95 \pm .04$ | $1.00 \pm .00$ | $.84 \pm .08$ | $.97 \pm .02$ |
| $\Diamond\,\ell_2^{\text{grad}}$ | $.99 \pm .01$ | $.98 \pm .00$ | $.81 \pm .09$ | $.95 \pm .03$ | $.43 \pm .20$ | $.84 \pm .06$ |

Table 6: Average success rate $\pm$ std. of each relevancy metric for top-10 identical subclass test. The metrics prefixed with $\diamondsuit$ are the ones we have repaired. The results with the average success rate over 0.5 are colored.

| | MNIST | | CIFAR10 | | | TREC | |
|---|---|---|---|---|---|---|---|
| Model | CNN | logreg | MobilenetV2 | CNN | logreg | Bi-LSTM | logreg |
| Parameter size | 12K | 8K | 2.2M | 12K | 31K | 20K | 7K |
| Accuracy | $0.99 \pm 0.00$ | $0.88 \pm 0.01$ | $0.92 \pm 0.01$ | $0.84 \pm 0.03$ | $0.71 \pm 0.03$ | $0.86 \pm 0.01$ | $0.81 \pm 0.02$ |
| $\ell_2^x$ | $.64 \pm .02$ | $.71 \pm .03$ | $.00 \pm .00$ | $.00 \pm .00$ | $.00 \pm .00$ | $.27 \pm .05$ | $.25 \pm .02$ |
| $\ell_2^{\text{last}}$ | $.54 \pm .04$ | - | $.00 \pm .00$ | $.00 \pm .00$ | - | $.30 \pm .02$ | - |
| $\ell_2^{\text{all}}$ | $.85 \pm .02$ | - | $.08 \pm .02$ | $.01 \pm .00$ | - | $.34 \pm .02$ | - |
| $\cos^x$ | $.67 \pm .02$ | $.74 \pm .03$ | $.00 \pm .00$ | $.01 \pm .01$ | $.00 \pm .00$ | $.28 \pm .04$ | $.27 \pm .02$ |
| $\cos^{\text{last}}$ | $.57 \pm .05$ | - | $.00 \pm .00$ | $.00 \pm .00$ | - | $.30 \pm .02$ | - |
| $\cos^{\text{all}}$ | $.89 \pm .02$ | - | $.16 \pm .02$ | $.02 \pm .01$ | - | $.34 \pm .02$ | - |
| $\text{dot}^x$ | $.21 \pm .02$ | $.23 \pm .03$ | $.00 \pm .00$ | $.00 \pm .00$ | $.00 \pm .00$ | $.05 \pm .01$ | $.05 \pm .01$ |
| $\text{dot}^{\text{last}}$ | $.08 \pm .02$ | - | $.00 \pm .00$ | $.00 \pm .00$ | - | $.14 \pm .01$ | - |
| $\text{dot}^{\text{all}}$ | $.79 \pm .03$ | - | $.13 \pm .02$ | $.01 \pm .01$ | - | $.17 \pm .02$ | - |
| IF | $.01 \pm .01$ | $.00 \pm .00$ | - | $.00 \pm .00$ | $.00 \pm .00$ | $.00 \pm .00$ | $.01 \pm .01$ |
| $\diamondsuit \ell_2^{\text{IF}}$ | $.14 \pm .03$ | $.16 \pm .02$ | - | $.00 \pm .00$ | $.00 \pm .00$ | $.11 \pm .02$ | $.24 \pm .02$ |
| $\diamondsuit \cos^{\text{IF}}$ | $.37 \pm .02$ | $.35 \pm .04$ | - | $.00 \pm .00$ | $.00 \pm .00$ | $.22 \pm .03$ | $.25 \pm .02$ |
| FK | $.00 \pm .00$ | $.00 \pm .00$ | - | $.00 \pm .00$ | $.00 \pm .00$ | $.00 \pm .00$ | $.00 \pm .00$ |
| $\diamondsuit \ell_2^{\text{FK}}$ | $.22 \pm .02$ | $.30 \pm .03$ | - | $.00 \pm .00$ | $.00 \pm .00$ | $.28 \pm .04$ | $.26 \pm .02$ |
| $\diamondsuit \cos^{\text{FK}}$ | $.58 \pm .02$ | $.46 \pm .04$ | - | $.00 \pm .00$ | $.00 \pm .00$ | $.41 \pm .03$ | $.25 \pm .02$ |
| GD | $.00 \pm .00$ | $.01 \pm .01$ | $.01 \pm .01$ | $.00 \pm .00$ | $.00 \pm .00$ | $.10 \pm .02$ | $.01 \pm .00$ |
| GC | $.86 \pm .03$ | $.87 \pm .02$ | $.06 \pm .02$ | $.01 \pm .01$ | $.01 \pm .01$ | $.37 \pm .03$ | $.37 \pm .02$ |
| $\diamondsuit \ell_2^{\text{grad}}$ | $.50 \pm .03$ | $.69 \pm .04$ | $.02 \pm .01$ | $.00 \pm .00$ | $.00 \pm .00$ | $.24 \pm .03$ | $.34 \pm .02$ |

| | AGNews | | Vehicle | | Segment | |
|---|---|---|---|---|---|---|
| Model | Bi-LSTM | logreg | MLP | logreg | MLP | logreg |
| Parameter size | 27K | 9K | 1K | 38 | 1K | 40 |
| Accuracy | $0.80 \pm 0.02$ | $0.80 \pm 0.01$ | $0.73 \pm 0.02$ | $0.73 \pm 0.01$ | $0.94 \pm 0.01$ | $0.90 \pm 0.01$ |
| $\ell_2^x$ | $.00 \pm .00$ | $.00 \pm .00$ | $.10 \pm .00$ | $.09 \pm .00$ | $.62 \pm .02$ | $.64 \pm .02$ |
| $\ell_2^{\text{last}}$ | $.01 \pm .00$ | - | $.07 \pm .00$ | - | $.66 \pm .07$ | - |
| $\ell_2^{\text{all}}$ | $.01 \pm .01$ | - | $.08 \pm .00$ | - | $.70 \pm .05$ | - |
| $\cos^x$ | $.01 \pm .00$ | $.02 \pm .01$ | $.10 \pm .00$ | $.09 \pm .00$ | $.46 \pm .02$ | $.48 \pm .02$ |
| $\cos^{\text{last}}$ | $.01 \pm .00$ | - | $.06 \pm .00$ | - | $.60 \pm .07$ | - |
| $\cos^{\text{all}}$ | $.02 \pm .01$ | - | $.10 \pm .00$ | - | $.62 \pm .08$ | - |
| $\text{dot}^x$ | $.01 \pm .00$ | $.02 \pm .01$ | $.00 \pm .00$ | $.00 \pm .00$ | $.24 \pm .02$ | $.25 \pm .02$ |
| $\text{dot}^{\text{last}}$ | $.01 \pm .00$ | - | $.00 \pm .00$ | - | $.01 \pm .02$ | - |
| $\text{dot}^{\text{all}}$ | $.02 \pm .00$ | - | $.00 \pm .00$ | - | $.02 \pm .05$ | - |
| IF | $.00 \pm .00$ | $.00 \pm .00$ | $.02 \pm .00$ | $.10 \pm .00$ | $.00 \pm .00$ | $.17 \pm .04$ |
| $\diamondsuit \ell_2^{\text{IF}}$ | $.01 \pm .00$ | $.10 \pm .01$ | $.02 \pm .00$ | $.20 \pm .00$ | $.15 \pm .11$ | $.17 \pm .08$ |
| $\diamondsuit \cos^{\text{IF}}$ | $.01 \pm .00$ | $.20 \pm .02$ | $.02 \pm .00$ | $.38 \pm .00$ | $.35 \pm .14$ | $.66 \pm .07$ |
| FK | $.00 \pm .00$ | $.00 \pm .00$ | $.02 \pm .00$ | $.00 \pm .00$ | $.00 \pm .00$ | $.00 \pm .00$ |
| $\diamondsuit \ell_2^{\text{FK}}$ | $.01 \pm .00$ | $.02 \pm .01$ | $.18 \pm .00$ | $.02 \pm .00$ | $.15 \pm .14$ | $.61 \pm .09$ |
| $\diamondsuit \cos^{\text{FK}}$ | $.01 \pm .00$ | $.09 \pm .01$ | $.10 \pm .00$ | $.01 \pm .00$ | $.53 \pm .09$ | $.75 \pm .07$ |
| GD | $.01 \pm .00$ | $.06 \pm .01$ | $.39 \pm .00$ | $.39 \pm .00$ | $.09 \pm .05$ | $.13 \pm .08$ |
| GC | $.04 \pm .00$ | $.10 \pm .01$ | $.16 \pm .00$ | $.30 \pm .00$ | $.52 \pm .09$ | $.64 \pm .04$ |
| $\diamondsuit \ell_2^{\text{grad}}$ | $.02 \pm .01$ | $.08 \pm .01$ | $.37 \pm .00$ | $.30 \pm .00$ | $.18 \pm .11$ | $.51 \pm .08$ |

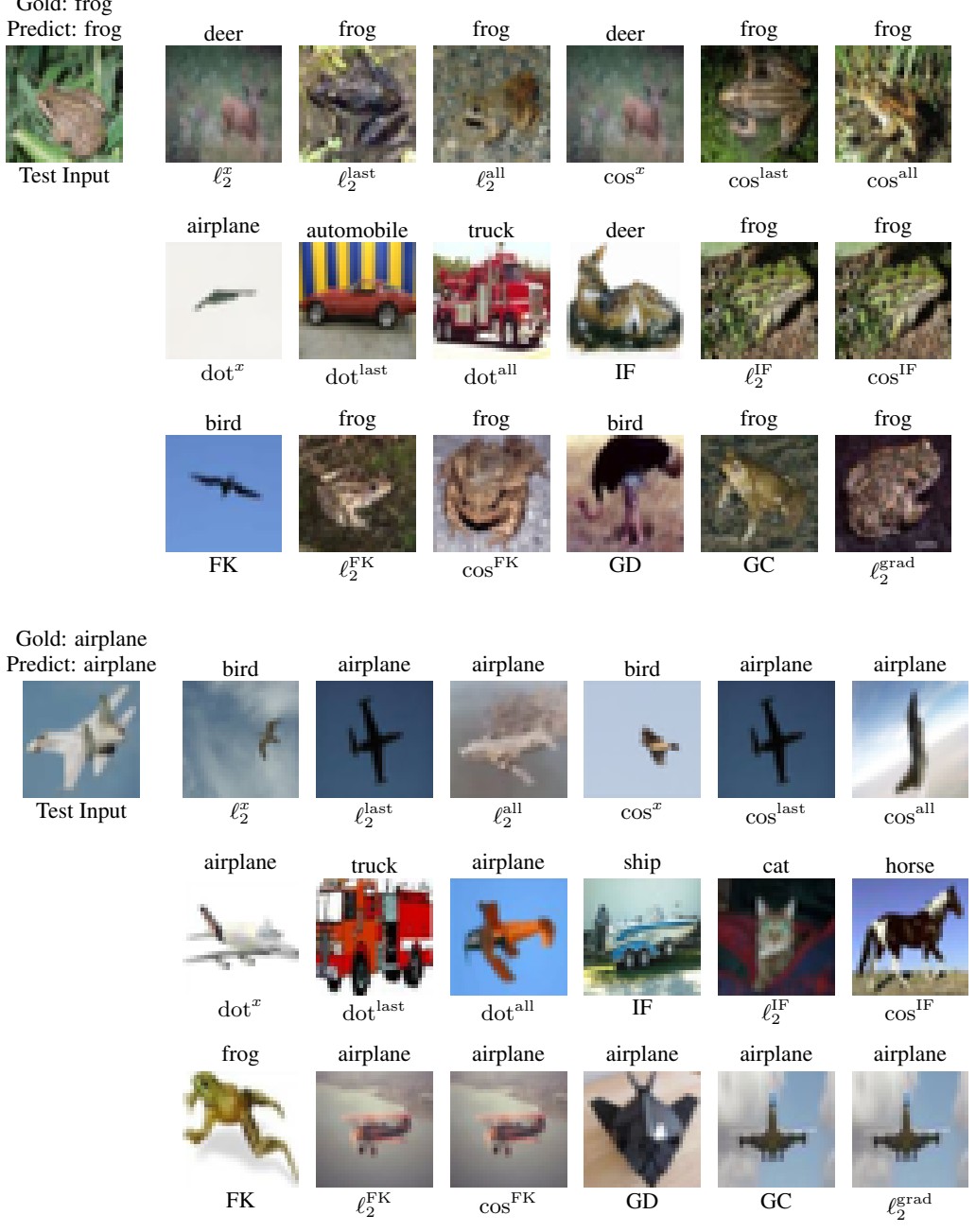

Figure 10: Relevant instances selected for random test inputs with correct prediction using several relevance metrics on CIFAR10 with CNN.

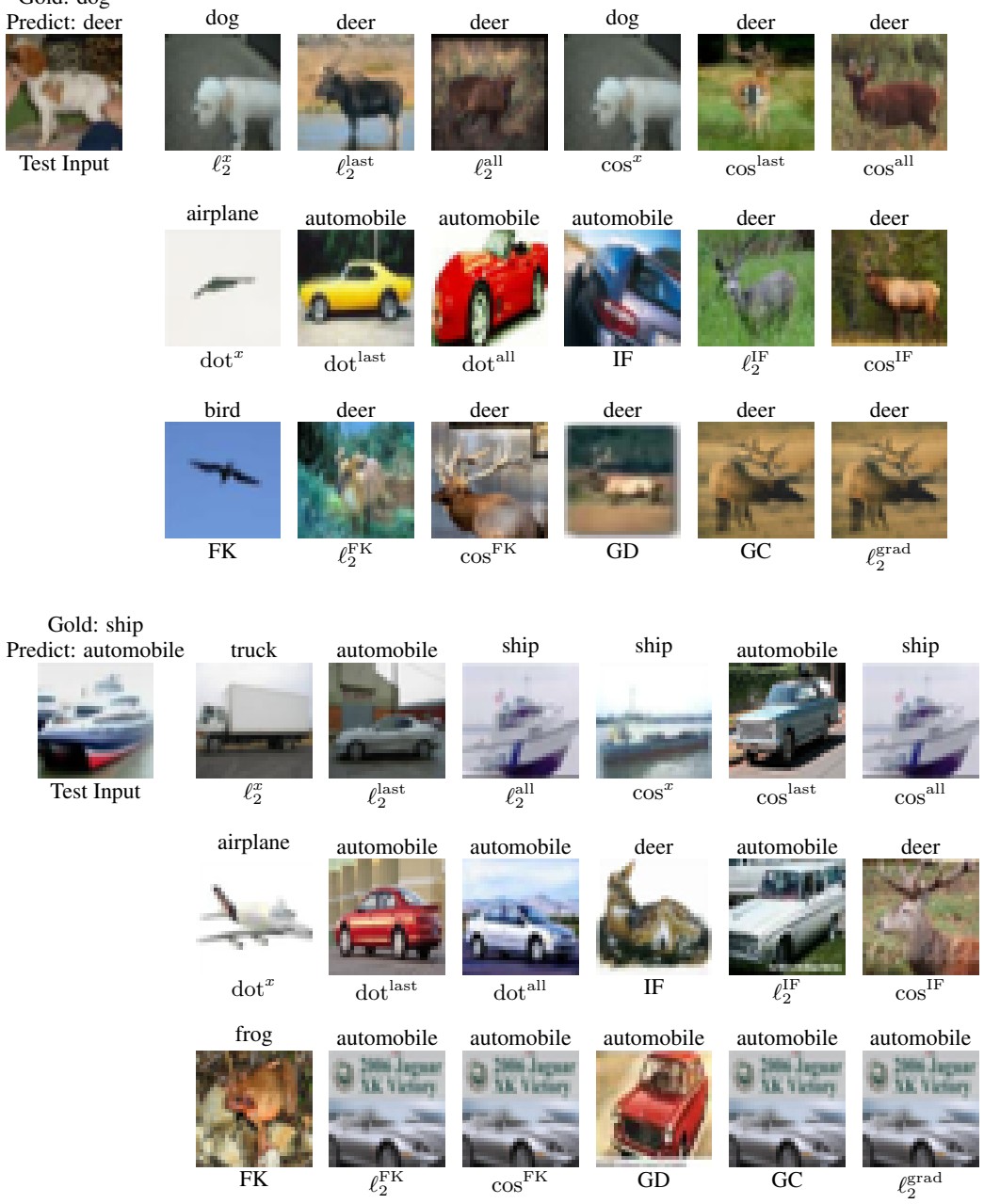

Figure 11: Relevant instances selected for random test inputs with incorrect prediction using several relevance metrics on CIFAR10 with CNN.

Table 7: Relevant instances selected for random test inputs with correct predictions using several relevance metrics on AGNews with LSTM. Out-of-vocabulary words are followed by [unk].

|  | Sentence | Class |
|---|---|---|
| Test Input | kerry widens lead in california , poll finds ( reuters ) | Gold: World
Predict: World |
| $\ell_2^x$ | in brief | Sci/Tech |
| $\ell_2^{\text{last}}$ | strong hurricane approaches[unk] bahamas[unk] , florida ( reuters ) | Sci/Tech |
| $\ell_2^{\text{all}}$ | strong hurricane approaches[unk] bahamas[unk] , florida ( reuters ) | Sci/Tech |
| $\cos^x$ | reuters poll : bush holds two - point lead over kerry ( reuters ) | World |
| $\cos^{\text{last}}$ | strong hurricane approaches[unk] bahamas[unk] , florida ( reuters ) | Sci/Tech |
| $\cos^{\text{all}}$ | strong hurricane approaches[unk] bahamas[unk] , florida ( reuters ) | Sci/Tech |
| $\text{dot}^x$ | reuters poll : bush holds two - point lead over kerry ( reuters ) | World |
| $\text{dot}^{\text{last}}$ | eurozone finance ministers debate action on oil as prices surge ( afp ) | World |
| $\text{dot}^{\text{all}}$ | business cash for bush campaign , lawyers[unk] for kerry ( reuters ) | World |
| IF | greek judoka[unk] dies in hospital after balcony[unk] suicide leap[unk] | Sports |
| $\ell_2^{\text{IF}}$ | world front | World |
| $\cos^{\text{IF}}$ | arafat family bickers[unk] over medical[unk] records of palestinian leader | World |
| FK | linux # 39;s latest moneymaker[unk] | Business |
| $\ell_2^{\text{FK}}$ | china launches zy-2[unk] resource[unk] satellite | Sci/Tech |
| $\cos^{\text{FK}}$ | china launches zy-2[unk] resource[unk] satellite | Sci/Tech |
| GD | judge adjourns[unk] ba[unk] # 39;asyir[unk] # 39;s trial until nov. 4 | World |
| GC | reuters poll : bush holds two - point lead over kerry ( reuters ) | World |
| $\ell_2^{\text{grad}}$ | reuters poll : bush holds two - point lead over kerry ( reuters ) | World |

|  | Sentence | Class |
|---|---|---|
| Test Input | some people not eligible[unk] to get in on google ipo | Gold: Sci/Tech
Predict: Sci/Tech |
| $\ell_2^x$ | insiders[unk] get rich[unk] through google ipo | Sci/Tech |
| $\ell_2^{\text{last}}$ | european judge probes microsoft antitrust case | Sci/Tech |
| $\ell_2^{\text{all}}$ | insiders[unk] get rich[unk] through google ipo | Sci/Tech |
| $\cos^x$ | insiders[unk] get rich[unk] through google ipo | Sci/Tech |
| $\cos^{\text{last}}$ | breakthrough in hydrogen[unk] fuel research | Sci/Tech |
| $\cos^{\text{all}}$ | insiders[unk] get rich[unk] through google ipo | Sci/Tech |
| $\text{dot}^x$ | italians[unk] , canadians[unk] gather[unk] to honour[unk] living legend[unk] : vc[unk] winner smoky[unk] smith[unk] ( canadian press ) | World |
| $\text{dot}^{\text{last}}$ | earnings alert : novell sees weakness[unk] in it spending | Sci/Tech |
| $\text{dot}^{\text{all}}$ | siemens backs new wireless technology | Sci/Tech |
| IF | matching[unk] wits[unk] on politics | Sports |
| $\ell_2^{\text{IF}}$ | insiders[unk] get rich[unk] through google ipo | Sci/Tech |
| $\cos^{\text{IF}}$ | congress probes fda in vioxx case | Business |
| FK | ' bin laden ' tape urges oil attack | Business |
| $\ell_2^{\text{FK}}$ | insiders[unk] get rich[unk] through google ipo | Sci/Tech |
| $\cos^{\text{FK}}$ | insiders[unk] get rich[unk] through google ipo | Sci/Tech |
| GD | issue 65 news hound[unk] : this week in gaming | Sci/Tech |
| GC | google responds[unk] to google news china controversy[unk] | Sci/Tech |
| $\ell_2^{\text{grad}}$ | insiders[unk] get rich[unk] through google ipo | Sci/Tech |

Table 8: Relevant instances selected for random test inputs with incorrect predictions using several relevance metrics on AGNews with LSTM. Out-of-vocabulary words are followed by [unk].

|  | Sentence | Class |
|---|---|---|
| Test Input | ibm to hire even[unk] more new workers | Gold:Sci/Tech Predict:Busi ness |
| $\ell_2^x$ | athletes[unk] to watch[unk] | Sports |
| $\ell_2^{last}$ | tech stocks tumble[unk] after chip makers warn | Business |
| $\ell_2^{all}$ | microsoft foe[unk] wins in settlement | Sci/Tech |
| $\cos^x$ | volkswagen[unk] workers stage new stoppages[unk] | Business |
| $\cos^{last}$ | tech stocks tumble[unk] after chip makers warn | Business |
| $\cos^{all}$ | microsoft revenue tops forecast | Business |
| $\text{dot}^x$ | italians[unk] , canadians[unk] gather[unk] to honour[unk] living legend[unk] : vc[unk] winner smoky[unk] smith[unk] ( canadian press ) | World |
| $\text{dot}^{last}$ | google up in market debut after bumpy[unk] ipo ( reuters ) | Business |
| $\text{dot}^{all}$ | google up in market debut after bumpy[unk] ipo ( reuters ) | Business |
| IF | greek judoka[unk] dies in hospital after balcony[unk] suicide leap[unk] | Sports |
| $\ell_2^{IF}$ | ibm # 39;s third - quarter earnings and revenue up | Business |
| $\cos^{IF}$ | arafat family bickers[unk] over medical[unk] records of palestinian leader | World |
| FK | great white sharks[unk] given new protection | World |
| $\ell_2^{FK}$ | ibm # 39;s third - quarter earnings and revenue up | Business |
| $\cos^{FK}$ | ibm to buy danish[unk] firms | Business |
| GD | some question speed of intel chief bill ( ap ) | World |
| GC | ibm shrugs[unk] off industry blues[unk] in q3 | Business |
| $\ell_2^{grad}$ | ibm # 39;s third - quarter earnings and revenue up | Business |

|  | Sentence | Class |
|---|---|---|
| Test Input | tougher[unk] rules wo n't soften[unk] law 's game | Gold: Sports Predict: Sci/Tech |
| $\ell_2^x$ | profiting[unk] from moore[unk] 's law | Business |
| $\ell_2^{last}$ | devil[unk] rays[unk] stuck[unk] in florida hours[unk] before game | Sports |
| $\ell_2^{all}$ | devil[unk] rays[unk] stuck[unk] in florida hours[unk] before game | Sports |
| $\cos^x$ | profiting[unk] from moore[unk] 's law | Business |
| $\cos^{last}$ | devil[unk] rays[unk] stuck[unk] in florida hours[unk] before game | Sports |
| $\cos^{all}$ | devil[unk] rays[unk] stuck[unk] in florida hours[unk] before game | Sports |
| $\text{dot}^x$ | italians[unk] , canadians[unk] gather[unk] to honour[unk] living legend[unk] : vc[unk] winner smoky[unk] smith[unk] ( canadian press ) | World |
| $\text{dot}^{last}$ | world 's top game players battle for cash ( ap ) | Sci/Tech |
| $\text{dot}^{all}$ | sportsnetwork[unk] game preview | Sports |
| IF | top grades[unk] rising again for gcses[unk] | World |
| $\ell_2^{IF}$ | calif. oks toughest[unk] auto emissions[unk] rules | World |
| $\cos^{IF}$ | un envoy headed to darfur | World |
| FK | yankee[unk] batters[unk] hit wall | Sports |
| $\ell_2^{FK}$ | a flat panel does n't always[unk] compute[unk] | Sci/Tech |
| $\cos^{FK}$ | a flat panel does n't always[unk] compute[unk] | Sci/Tech |
| GD | issue 65 news hound[unk] : this week in gaming | Sci/Tech |
| GC | atari[unk] announces first 64-bit[unk] game | Sci/Tech |
| $\ell_2^{grad}$ | atari[unk] announces first 64-bit[unk] game | Sci/Tech |

