# OpenReview forum: "Evaluation of Similarity-based Explanations"
_ICLR.cc/2021/Conference — ICLR 2021 Poster_

### Official Review · AnonReviewer1 · 2020-10-22
**Evaluation of similarity-based explanations**

**Rating:** 6
**Confidence:** 3

**Review:**

Summary:


This paper is on the evaluation of XAI for supervised models. XAI is a trending topic since several years now but evaluating the quality of an explanation is still challenging. This works focuses on similarity-based explanations, which means providing similar training examples along with the predicted class. Another assumption is that the chosen examples must be of the same class.

The main contribution is to benchmark several similarity measures against various datasets, using three tests. One test is taken from the literature and the two others are proposed in the paper. The experiments show that some similarity measures are better than others. I really like that the authors find an explanation for those results, based on geometrical properties of the measured objects. Besides, they use this finding to correct existing measures.


##########################################################################

Reasons for score:

As a whole, I like the main aim of the paper for I think XAI deserves better evaluation metrics. However, I have some concerns that explains my vote.

First of all, I am not fully convinced that the assumption of restricting to similar examples with the *same* class is really good for giving a relevant explanation. I believe that a good explanation will need both positive and negative examples, in particular near the decision frontier. Therefore this tempers the interest of the proposal made in this paper.

Second, I had a hard time figuring out how and when the third test (with subclasses) can be used, and how it has been implemented in the presented experiments. From my understanding, this test is useful when you access a class taxonomy (e.g., animal -> mammal -> cat). But in Section 4.2 the authors explain that they randomly assign existing classes to on of the two (subclass) A and B. What does it mean? Besides I'm not sure to be convinced by the argument raised in 3.3 ("...the violation of this third requirement again leads to nonsense explanation"). We can find similarities between two animals (here, car and frog) because they share common properties (e.g., breathing), in particular if you compare them to a truck.

Third, the randomization test relies on a unique random model that is confronted to the learned model. However it is possible that the random model is close to the learned model and I expected the random model to be generated several times in order to get more robust results.

Finally, I was puzzled by Fig.6. Why don't the authors use the same examples for the different measures? It seems unfair as the examples given to GC looks easier to classify/explain.


##########################################################################Pros:


1. address the difficult task of evaluating explanations in XAI
2. paper easy to read (except for the third test)
3. really interesting discussion on the failure of (some) existing measures

##########################################################################

Cons:


1. restricted to a narrow form of XAI
2. subclass test not well explained
3. Fig.6 is misleading (see above)


##########################################################################

Questions during rebuttal period:

Please address and clarify the cons above.


#########################################################################

Additionnal comments:

"two models with different reasoning process": this expression is misleading, it's not really a "reasoning process". I would write "inference process" or "inductive process".

The accuracy of the learned models is never taken into account. I think evaluating the explanations should be done at the light of classification ability. The authors may discuss this point.

Some typos:

"two instance"
"the the" (twice)

---

> ### Author Response · Authors · 2020-11-17
> **Response to Reviewer 1**
>
> We thank the reviewer for the comments. Please see below for our response to each comment.
>
> **restricted to a narrow form of XAI.**
> **a good explanation will need both positive and negative examples, in particular near the decision frontier.**
> We totally agree that explanation methods other than similarity-based explanations, such as providing negative examples, are also important for practical purposes. However, it would not be reasonable to evaluate different types of explanations on the same metric. Therefore, we narrowed our focus to similarity-based explanations.  The methods for providing positive and negative examples are designed for different purposes, and there is no need to evaluate them on the same test. They should be evaluated individually according to the purpose of the explanation. The evaluations of other types of instance-based explanations is beyond the scope of the paper: designing evaluation criteria for other types of explanations will be an important future direction of the field. We believe that this study to be a first step towards this direction.
>
> **subclass test not well explained**
> We would like to remind that we adopted the identical subclass test as a minimal requirement for the explanation that is plausible to any users. As pointed out, we agree that some users will find frog to be an appropriate explanation for a cat being animal, by inferring the taxonomy of the classes. However, such an inference may differ across the users, and we cannot hope of all the users to make the same inference. When the users failed to infer the taxonomy, the users will find the explanation implausible. To make the explanation plausible to any users, the only choice is to provide instances of the same subclass. We will clarify the point above in the manuscript.
>
> **randomly assign existing classes to on of the two (subclass) A and B**
> As we mentioned above, we consider the explanations that are irrelevant to the class taxonomy are ideal. Our experiment is designed so that the underlying taxonomy to not affect the capability of each relevance metric. For example, some metric might perform well/worse when a certain taxonomy exists. We tried to remove such effects that might present in limited applications, and we intended to measure the general performance of each metric independent of the underlying taxonomy.
>
> **the randomization test should be done several times**
> We performed all tests, including the randomization test, ten times with a different random seed. We reported these averages and standard deviations in appendix F.
>
> **Fig.6 is misleading**
> Please remind that what we have shown in Fig.6 are not typical examples of the each explanation method. What Fig.6 shows are "the training instances selected as most relevant for multiple test instances" in each method. We used Fig.6 for explaining the reasons of the failure of some methods in terms of their norms and angles of the gradients. We cannot choose the same example because "the training instances selected as most relevant for multiple test instances" are different for each method. GC may seem easy because they could provide good explanations by ignoring the effect of the norms (see Sec.5). We never cherry picked the examples.
> As a response to the request for providing typical examples, we have added some examples obtained by each method for random test instances to Appendix G.

---

> > ### Comment · AnonReviewer1 · 2020-11-18
> > **ok**
> >
> > Thank you for your answers that help me clarify the interest of your contribution.  Even though I still continue to think that it is rather limited in term of XAI, I will slightly increase my rating.

---

> > > ### Author Response · Authors · 2020-11-20
> > > **Thank you for your understanding**
> > >
> > > Thank you for your understanding of our work. Let us add just one more point about the narrowness of our scope you pointed out.
> > > Explanation using similar examples is one important class of explanation methods, as some literature has considered. For example, [Ref1:Sec6], [Ref2:Sec3.2.4], and [Ref3:Sec1.1] raise several real situations where humans explain the decisions by raising similar examples. This proves the importance and effectiveness of this type of explanation.
> > > We thus believe that our study makes a rigid contribution to an important class of explanation methods.
> > >
> > > [Ref1] Christoph Molnar. 2018.Interpretable Machine Learning: A Guide for Making Black Box Models Explainable.  https://christophm.github.io/interpretable-ml-book/
> > > [Ref2] Zachary C. Lipton. 2017. The Mythos of Model Interpretability. arXiv preprint arXiv:1606.03490.
> > > [Ref3] Agnar Aamodt, Enric Plaza. 1994. Case-Based Reasoning: Foundational Issues, Methodological Variations, and System Approaches. Artificial Intelligence Communications.

---

### Official Review · AnonReviewer4 · 2020-10-30
**The study reported an systematic evaluation of existing metrics used in generating similarity-based explanations. Overall, the paper is technically sound and easy to understand.**

**Rating:** 7
**Confidence:** 4

**Review:**

The presented study tackled the problem of evaluating different metrics for generating similarity-based explanations. To this end, the authors used three tests to evaluate four types of metrics on several different datasets. The experiment results revealed that the cosine similarity of the gradients of the loss performed the best. Overall, the methods adopted by the study are technically sound and the paper is well written (though with a few typo errors). In particular, the presented experiments are extensive and the results are somewhat interesting. Still, there are a few comments that can be considered to further improve the paper:

1. It would be good to provide a more comprehensive description of relevant studies in generating explanations (e.g., what are those application scenarios, what datasets were used).
2. Why only logistic regression and deep neural networks like CNN and Bi-LSTM were used for experiments? Any justification for this choice?
3. It would be good to provide some examples (especially those by using textual data) to provide a more in-depth explanation the actual value behind those reported numbers, e.g., what specific explanations can be produced by applying the best-performing metric vs. the worst-performing metric?

---

> ### Author Response · Authors · 2020-11-17
> **Response to Reviewer 4**
>
> We thank the reviewer for the comments. Please see below for our response to each comment.
>
> **It would be good to provide a more comprehensive description of relevant studies in generating explanations**
> We provided a brief summary of the related studies is given in section 1.2, as well as the details of the methods in appendix A. In terms of the applications and datasets, for example, IF [Koh & Liang, 2017] was used for (i) instance-based explanation on the dog vs. fish image classification dataset, and (ii) data cleansing on email spam classification dataset. An in-depth review of individual studies is beyond the scope of the current paper. We suggest that interested readers to see the individual paper.
>
> **Why only logistic regression and deep neural networks like CNN and Bi-LSTM were used for experiments?**
> DNN is the highest-performing in many tasks, and we considered that it is a natural choice. Since some methods, such as IF, do not have theoretical justification in non-convex settings such as DNN, we also used linear logistic regression. Kernel logistic regression/SVM are other possible choices. However, investigating them will only reveal the effect of the different basis function (note that the linear logistic regression is a special case with the linear kernel), so we believe that an investigation of standard logistic regression is sufficient. Also, most explanation methods assume differentiable functions, so non-differentiable models such as decision trees and random forests are out of scope.
>
> **It would be good to provide some examples**
> We agree with your suggestion. We added some examples to the appendix G. Please see it.

---

### Official Review · AnonReviewer2 · 2020-10-31
**Empirical comparative study of metrics for explanations with simple requirements.**

**Rating:** 6
**Confidence:** 3

**Review:**

The authors investigate which relevance metrics are desirable for explanations that are based on extracting similar instances as evidence to support a model prediction. They establish three minimal requirements for those metrics:  one adapted from the state-of the art: (1) model randomization to ensure the measure is model-specific, and two proposed requirements,  (2) identical class test (instance pointed out should be of the same class as the test instance) and (3) identical subclass test (instance pointed out should be of the same subclass as test instance). Several metrics are compared and the authors conclude that many of the state-of-the-art metrics do not even satisfy the first requirement and that the cosine similarity of the gradients of the  loss perform best.
The authors conduct an empirical evaluation on two image datasets, two text datasets and one tabular dataset.
The paper is well-written, the method and the two requirements straightforward and the empirical comparative approach acceptable.
It is not clear from the paper how the noise in the labels is handled (what if the nearest neighbor used in the second requirement is mislabeled?). More extensive experiments on tabular data would have been good.
Furthermore, how to ensure these three requirements are sufficient?  While the approach presented and the two additional principles are quite simple, it is nice to see a comparative study on explanation metrics. The work could benefit from deeper insights on the evaluation metrics from a theoretical perspective.

Minor comments:
- Figure 1 and 2 are not necessary in my opinion
- Figure 3: sub-captions. missing.
- concrete examples on the vehicle data would have been desirable. Why this data specifically? there are many benchmark datasets from the UCI repository and others.
- "do" --> does in section 3
- "the the" in section 3.1
- "do" -> does on section 3.2

---

> ### Author Response · Authors · 2020-11-17
> **Response to Reviewer 2**
>
> We thank the reviewer for the comments. Please see below for our response to each comment.
>
> **It is not clear from the paper how the noise in the labels is handled.**
> We agree that an explanation with mislabeled instances will not be plausible to the users. However, it is the problem of the trained model, and it is not the fault of the relevance metric. Because our target is evaluating the validity of the relevance metrics, we do not count the fault of the model as the failure of the relevance metric. We therefore think that no special treatment is necessary for label noise.
> Suppose a test image of cat is classified as cat. There are two types of label noises to be considered in this case: (i) a training image of cat is labeled as something else (e.g., dog), and (ii) a training image whose true class is other than cat (e.g., a dog image) is labeled as cat. From the perspective of the trained model, there is no reason to discriminate the mislabeled training images from the other correctly labeled images (because the model is trained in that way). That is, the model should consider the image (i) as dog equally as the other images of dog, and the image (ii) as cat equally as the other images of cat. Any faithful explanations that reflect this model’s inference process would result in implausible explanation to the users because the model itself is not ideal but not because the relevance metric is corrupt.
>
> **More extensive experiments on tabular data would have been good.**
> We agree with the comment. We added the results to the appendix F. Please see it.
>
> **how to ensure these three requirements are sufficient?**
> We do not consider that these requirements are sufficient. We designed these tests to check the minimum requirements (see Sec.1 and Sec.3). In other words, the purpose of these tests is to find methods that do not meet the minimal requirements for the practical usage.
> Note that, even with these three tests, the current results indicate that most of the relevance metrics do not to meet the minimal requirements for the practical usage. This observation suggests that the development of relevance metrics that meet these minimal requirements is essential.
> Designing further evaluation criteria is, of course, an important future direction of the field. We believe that this study to be a first step towards this direction.
>
> **it is nice to see a comparative study on explanation metrics.**
> To the best of our knowledge, there is no well-known evaluation criteria for similarity-based explanations. We are happy to add comparisons if there are any criteria that we are not aware of. Could you raise a few relevant papers if you know any?
>
> **The work could benefit from deeper insights on the evaluation metrics from a theoretical perspective.**
> We totally agree with this point. We are happy if you can give us some pointers to the studies that will be helpful for the theoretical analysis.

---

### Official Review · AnonReviewer3 · 2020-11-02
**Interesting motivation but the correctness of the evaluation criteria is unclear. More discussion needed**

**Rating:** 5
**Confidence:** 4

**Review:**

#### Summary:
 This work provides an empirical evaluation of similarity metrics used in example-based explanations methods, where the goal is to provide decision support examples in the training set for a black-box model's prediction. The paper evaluates gradient based metrics popular in the literature such as Influence functions, fisher kernels but also simpler naive approaches that relies on l2, cosine distances and dot product on different embedding spaces. The authors introduce 2 new tasks for assessing the reliability of the different methods: an identical class test and an identical subclass test.

#### Strengths
 - The paper is clearly written and easy to follow. The idea to empirically evaluate different similarity metrics for example-based explanation methods is interesting and could be useful to practitioners. The motivation for the work is clear and well introduced.

#### Weaknesses
- Identical class test: the requirement that the returned explanatory examples in the training set should belong to the same class as the example is ambiguous and raises some concerns.  1) Is is not clear how this applies in case where the test example is misclassified? Why does the identity requirement applied only on the predicted label and not the true label as well? 2) Training examples that are either predicted differently or belong to a different true class as the test example can be useful in understanding failure modes of the model, misclassified examples, or inhibitory examples. This validity of this test need to be discussed and justified.
- Same applies for the Identity subclass test. The subclass test has even stronger requirements than the identity class test. The validity of this test should be justified beyond a simple definition.

-Strong statements are made on the 'usefulness' of the explanations but these are mainly intuitive assessment of individual cases.  For example, in section 3.2, the authors state "The violation of this requirement leads to nonsense explanations such as “I think this image is cat because a similar image I saw in the past was dog.” which do not make sense to the users at all. When faced with such an explanation, the users will find the validity of the model’s prediction questionable, and will ignore the prediction even if the underlying model is valid".
The usefulness of the explanations provided by these methods to human users should be assessed through a human evaluation.

#### Recommendation with reasons
While I like to motivation behind this work and it's potential usefulness, the validity of the proposed tests is not clearly addressed. For that reason, I can not recommend acceptance for this work.

#### Questions
- Justify the correctness of the identity and subclass identity test

#### Additional feedback
Additional reference: Yeh, C. K., Kim, J., Yen, I. E. H., & Ravikumar, P. K. (2018). Representer point selection for explaining deep neural networks. In Advances in Neural Information Processing Systems (pp. 9291-9301)

---

> ### Author Response · Authors · 2020-11-17
> **Response to Reviewer 3**
>
> We thank the reviewer for the comments. Please see below for our reply.
>
> **Additional reference**
> We would like to appreciate your kind effort for raising a relevant paper although it is already cited in the paper.
>
> **Correctness of the tests**
> A major concern of the reviewer is the correctness of the two tests for plausibility and the need for human evaluation.
> First of all, we would like to emphasize that our focus is on explanations in the format of "This image is cat because this similar image is cat." There will be no doubt that explanations that do not satisfy the requirements are logically broken (e.g., "This image is cat because this similar image is frog.").
> Our tests will be invalid only if there are some situations where a logically broken explanation is helpful. However, we think that this will not likely be the case. We raise [Ref1] as a supporting evidence. In [Ref1], the authors reveled a bias in humans, called algorithm aversion, that people tend to ignore the algorithm if it makes errors. Because of this bias, we believe that providing a logically broken explanation will make the users to ignore the model.
>
> Below, we reply to two specific concerns about the identical class test.
>
> **(1) Why does the identity requirement applied only on the predicted label and not the true label as well?**
> We focus on explanations in the format of "This image is cat because this similar image is cat." Consider an example where a test image of cat is misclassified as frog. If the requirement is applied to the true label, the explanation "This image is frog because this similar image is cat." is allowed, although this explanation is logically corrupt. We therefore apply the identical class requirement only to the predicted label.
>
> **(2)Training examples that are either predicted differently or belong to a different true class as the test example can be useful. The validity of this test need to be discussed and justified.**
> We totally agree that some instances in other classes are also helpful in practice. However, such explanations are outside of the scope of the current paper. Please remind that we do not claim the validity of our tests for the explanations in the format other than "This image is cat because this similar image is cat.” For example, it is apparent that the identical class/subclass tests are not appropriate for evaluating the explanations that raises instances in other classes. For explanations in such other formats, we need to design evaluation criteria for them which is beyond the scope of the current paper.
>
> [Ref1] Dietvorst, Berkeley J et al. 2015. Algorithm aversion: people erroneously avoid algorithms after seeing them err. Journal of experimental psychology. General vol. 144(1): 114-126.

---

### Decision · Program_Chairs · 2021-01-07
**Final Decision**

**Decision:**

Accept (Poster)

**Comment:**

This paper performs an empirical comparison of similarity-based attribution methods, which aim to "explain" model predictions via training samples. To this end, the authors propose a handful of metrics intended to measure the acceptability of such methods. While one reviewer took issue with the proposed criteria, the general consensus amongst reviewers is that this provides at least a start for measuring and comparing instance-attribution methods.

In sum, this is a worthwhile contribution to the interpretability literature that provides measures for comparing and contrasting explanation-by-training-example methods.